# On the Interplay between Social Welfare and Tractability of Equilibria

**Ioannis Anagnostides**
Carnegie Mellon University
`ianagnos@cs.cmu.edu`

**Tuomas Sandholm**
Carnegie Mellon University
Strategic Machine, Inc.
Strategy Robot, Inc.
Optimized Markets, Inc.
`sandholm@cs.cmu.edu`

## Abstract

Computational tractability and social welfare (aka. efficiency) of equilibria are two fundamental but in general orthogonal considerations in algorithmic game theory. Nevertheless, we show that when (approximate) *full efficiency* can be guaranteed via a *smoothness* argument à la Roughgarden, Nash equilibria are approachable under a family of no-regret learning algorithms, thereby enabling fast and decentralized computation. We leverage this connection to obtain new convergence results in *large games*—wherein the number of players $n \gg 1$—under the well-documented property of full efficiency via smoothness in the limit. Surprisingly, our framework unifies equilibrium computation in disparate classes of problems including games with vanishing *strategic sensitivity* and two-player zero-sum games, illuminating en route an immediate but overlooked equivalence between smoothness and a well-studied condition in the optimization literature known as the *Minty property*. Finally, we establish that a family of no-regret dynamics attains a welfare bound that improves over the smoothness framework while at the same time guaranteeing convergence to the set of coarse correlated equilibria. We show this by employing the *clairvoyant* mirror descent algortihm recently introduced by Piliouras *et al.*

## 1 Introduction

The *Nash equilibrium (NE)* [76] formalizes the notion of a *stable* outcome in a multiagent strategic interaction, and has arguably served as the most influential solution concept in the development of game theory. Indeed, algorithms designed to approximate Nash equilibria in two-player zero-sum games have recently resolved major challenges in AI [7, 10, 81]. Its prescriptive power, however, has been severely undermined in multi-player general-sum games by intrinsic computational barriers [30, 20, 94, 35, 28, 44], limitations which also manifest in the inability of natural learning algorithms to converge [73, 105, 60, 70]. Another well-established but orthogonal critique is the *equilibrium selection* problem [52]: a general-sum game may have multiple Nash equilibria with widely different welfare. As a result, a modern research agenda in computational game theory has been to identify and characterize natural classes of games that circumvent those fundamental limitations.

In this paper, we uncover a new natural class of games for which the aforementioned caveats of Nash equilibria can be effectively addressed. In particular, our investigation originates from a natural question: when are *efficient*—in terms of social welfare—Nash equilibria easy to compute? The answer to this question is, at first glance, unsatisfactory: even if Nash equilibria are fully efficient, computational hardness still persists given that constant-sum multi-player games are hard. Indeed, efficiency and computational tractability are, in general, two orthogonal considerations. We show,

37th Conference on Neural Information Processing Systems (NeurIPS 2023).

however, an interesting twist, an unexpected interplay between efficiency—when viewed from a specific lens—and the behavior of a family of *no-regret* learning algorithms.

## 1.1 Our results

To elucidate the alluded connection that drives much of our results, we first have to recall that the canonical paradigm for establishing the efficiency of equilibria is Roughgarden's celebrated *smoothness* framework [90] (exposed thoroughly in Section 2). In this context, we observe that *if full efficiency of equilibria can be guaranteed via a smoothness argument (with bounded parameters), then Nash equilibria are approachable under a family of no-regret learning algorithms.* In other words, full efficiency *via smoothness* implies computational tractability of NE. This is surprising in that Roughgarden's smoothness framework was developed primarily in order to automatically extend *price of anarchy (PoA)* bounds to more permissive equilibrium concepts, such as *coarse correlated equilibria (CCE)*; tractability of NE appears at first glance entirely unrelated. In fact, while applying the smoothness framework to (multi-player) constant-sum games appears to make little sense, given that PoA considerations are trivial in such games (all outcomes attain the same social welfare), it turns out that a certain regime of smoothness in (multi-player) constant-sum games is equivalent to a well-known condition in the optimization literature called the *Minty property* [36] (Observation 3.4).

The condition described above already captures—somewhat unexpectedly—well-studied settings, such as games that admit a minimax theorem [13], but we have found that the most fertile and novel ground to apply this theory revolves around *large games*, that is, games with a large number of players $n \gg 1$.[1] The reason we focus on large games is a well-documented economic phenomenon: equilibria in large games approach—under natural conditions—full efficiency as $n \to +\infty$, a property often established via smoothness [39, 23]—a crucial ingredient in our framework. (One rough intuition for this is that in large games each player's influence on the outcome becomes negligible, making optimal behavior easier to characterize [39].) To state our first main result, we denote by $(\lambda, \mu)$ a pair of bounded *smoothness* parameters of a game $\mathcal{G}$; the ratio $\rho(\lambda, \mu) \coloneqq \frac{\lambda}{1+\mu}$ circumscribes the (in)efficiency of equilibria of $\mathcal{G}$, in that all equilibria will attain at least a $\rho$ fraction of the optimal welfare.

**Theorem 1.1** (Informal). *Consider a sequence of $n$-player $(\lambda_n, \mu_n)$-smooth games $(\mathcal{G}_n)_{n \geq 1}$ such that $\rho_n \coloneqq \frac{\lambda_n}{1+\mu_n} \to 1$ with a sufficiently fast rate. Then, there are decentralized and computationally efficient no-regret dynamics approaching an $(o_n(1), o_n(1))$-weak Nash equilibrium.*

A few remarks are in order. First, in Theorems 3.1 and A.2 we give a precise non-asymptotic characterization that quantifies the number of iterations as well as the approximation error. We also recall that in an $(o_n(1), o_n(1))$-*weak* Nash equilibrium *almost all* players are *almost best responding* (Definition 2.2); this is a well-studied relaxation for which hardness results carry over in general [4, 2, 5, 94]. A limitation of a weak Nash equilibrium is that it can prescribe a strategy profile in which a large numbers of players—albeit of vanishing fraction—can significantly benefit from deviating; this limitation is inherent in a certain regime of Theorem 1.1. Depending on the rate with which $\rho_n$ approaches full efficiency, Theorem 1.1 can also imply convergence to the usual notion of Nash equilibrium—wherein *all* players are almost best responding (Corollary A.5). More generally, for a broad class of games that includes *graphical* games with bounded degree, *polymatrix* games, and games exhibiting small *strategic sensitivity*, the conclusion of Theorem 1.1 applies without imposing any restrictions on the rate of convergence of $\rho_n$; those are the most commonly studied classes of games when $n \gg 1$.

There are ample compelling aspects in connecting the convergence of no-regret learning algorithms with Roughgarden's smoothness framework. First, there has been a considerable interest in understanding smoothness, insights that can now be inherited to a seemingly entirely different but equally fundamental problem. For example, smoothness naturally extends to *Bayesian games* [101, 91, 53] (see also [97] for the related class of contextual games), a property that we leverage in Theorem 3.8 to expand our scope to mechanisms of incomplete information. Additional extensions based on *local smoothness* [92, 78] and the refined primal-dual framework of Nadav and Roughgarden [75] are also described in Appendices A.3 and A.4. Finally, our criterion has a clear and natural economic interpretation, and is part of an ongoing research effort to identify tractable classes of variational inequalities (VIs) beyond the Minty property [33].

---

[1]We use the terminology large games here—in accordance with much of the economics literature—to refer to games with a large number of players; it should not be confused with another common usage referring to games with a "large" action space.

As a concrete example, our framework subsumes games wherein each player's effect on the outcome vanishes as $n \to \infty$; this captures, for example, simple voting settings [57], as well as general auction design problems where establishing that property turns out to be highly non-trivial and quite delicate [39]. More concretely, for games with vanishing *sensitivity* $\epsilon_n \to 0$, in that unilateral deviations can only affect a player's utility by an additive $\epsilon_n$ (Definition 3.5), we show that the conclusion of Theorem 1.1 applies as long as $\epsilon_n \le o_n(1/\sqrt{n})$ (Theorem 3.6). At the same time, the condition that $\rho_n \to 1$ goes much deeper than games with vanishing sensitivity, and, as we explained earlier, surprisingly applies to two-player zero-sum games as well. We find it conceptually appealing that a unifying framework can establish tractability of Nash equilibria in two seemingly disparate classes of problems such as two-player zero-sum games and games with vanishing strategic sensitivity.

Remaining on smooth games, but relaxing the assumption $\rho \approx 1$, we next study conditions under which the efficiency guaranteed by the smoothness framework can be improved, while at the same time ensuring the no-regret property, thereby implying convergence to the set of CCE. The smoothness bound is known to be applicable to *any* outcome of no-regret dynamics, but here we are instead interested in more refined guarantees when specific learning algorithms are in place. Building on a recent result [1], we show in Theorem 4.1 that the *clairvoyant* variant of gradient descent, introduced by Piliouras et al. [85], enjoys an improved welfare bound *and* ensures fast convergence to the set of CCE for the average correlated distribution of play. Crucially, compared to an earlier result [1], the clairvoyant algorithm manages to satisfy an appealing notion of per-player incentive compatibility, in the form of convergence to CCE. In other words, improving the welfare predicted by smoothness is not at odds with incentive compatibility (Corollary 4.3).

## 2   Background

**Notation**   We let $\mathbb{N} = \{1, 2, \dots\}$ be the set of natural numbers. For $n \in \mathbb{N}$, we denote by $[\![n]\!] := \{1, 2, \dots, n\}$. For a vector $\boldsymbol{x} \in \mathbb{R}^d$, we denote by $\|\boldsymbol{x}\|_2$ its (Euclidean) $\ell_2$ norm. For a convex, compact and nonempty set $\mathcal{X}$, we let $\mathcal{P}_{\mathcal{X}}(\cdot)$ represent the Euclidean projection operator with respect to $\mathcal{X}$. We let $D_{\mathcal{X}}$ be the $\ell_2$-diameter of $\mathcal{X}$. To simplify the exposition, we often use the $O(\cdot)$ notation in the main body to suppress the dependence on certain parameters; we also write $O_n(\cdot)$ to indicate the asymptotic growth solely as a function of $n$, so as to lighten the exposition.

**Multilinear games**   We consider $n$-player *multilinear* games. In a multilinear game $\mathcal{G}$ each player $i \in [\![n]\!]$ has a convex, compact and nonempty set of feasible strategies $\mathcal{X}_i \subseteq \mathbb{R}^{d_i}$, for some dimension $d_i \in \mathbb{N}$. Under a joint strategy profile $\boldsymbol{x} = (\boldsymbol{x}_1, \dots, \boldsymbol{x}_n) \in \prod_{i=1}^n \mathcal{X}_i =: \mathcal{X}$, there is a continuous utility function $u_i : (\boldsymbol{x}_1, \dots, \boldsymbol{x}_n) \mapsto \mathbb{R}$ such that $u_i(\boldsymbol{x}) = \langle \boldsymbol{x}_i, \boldsymbol{u}_i(\boldsymbol{x}_{-i}) \rangle$, for some function $\boldsymbol{u}_i : \boldsymbol{x}_{-i} \mapsto \mathbb{R}^{d_i}$; here, we used for convenience the standard notation $\boldsymbol{x}_{-i} := (\boldsymbol{x}_1, \dots, \boldsymbol{x}_{i-1}, \boldsymbol{x}_{i+1}, \dots, \boldsymbol{x}_n)$. This setup readily captures *normal-* as well as *extensive-form* games.

Specifically, in a normal-form game every player $i \in [\![n]\!]$ selects as strategy a probability distribution $\boldsymbol{x}_i \in \Delta(\mathcal{A}_i)$ over a finite set of available actions $\mathcal{A}_i$. There is an arbitrary utility function $u_i : \mathcal{A} := \prod_{i=1}^n \mathcal{A}_i \to [-1, 1]$ that maps a joint action profile to a utility for Player $i$; we will be making the standard assumption that the range of each player's utilities is bounded by an absolute constant, which is in particular independent of the number of players $n$. In this setting, the mixed extension of the utility function indeed satisfies the multilinearity condition imposed above: for $\boldsymbol{x} \in \prod_{i=1}^n \Delta(\mathcal{A}_i)$, $u_i(\boldsymbol{x}) := \mathbb{E}_{\boldsymbol{a} \sim \boldsymbol{x}}[u_i(\boldsymbol{a})] = \langle \boldsymbol{x}_i, \nabla_{\boldsymbol{x}_i} u_i(\boldsymbol{x}) \rangle$, where we overloaded the notation $u_i(\cdot)$.

Returning to the general setting of multilinear games, we let $F_{\mathcal{G}} : \prod_{i=1}^n \mathcal{X}_i \to \prod_{i=1}^n \mathcal{X}_i$ denote the underlying *operator* of the game $\mathcal{G}$, defined as $F_{\mathcal{G}} : (\boldsymbol{x}_1, \dots, \boldsymbol{x}_n) \mapsto (\boldsymbol{u}_1(\boldsymbol{x}_{-1}), \dots, \boldsymbol{u}_n(\boldsymbol{x}_{-n}))$, where function $\boldsymbol{u}_i$ was introduced earlier. For notational simplicity, we will often omit the subscript $\mathcal{G}$ when it is clear from the context. We will say that $F$ is *L-Lipschitz continuous* (w.r.t. the $\ell_2$ norm) if for any $\boldsymbol{x}, \boldsymbol{x}' \in \prod_{i=1}^n \mathcal{X}_i$ it holds that $\|F(\boldsymbol{x}) - F(\boldsymbol{x}')\|_2 \le L\|\boldsymbol{x} - \boldsymbol{x}'\|_2$.

**Welfare and the price of anarchy**   For a joint strategy profile $\boldsymbol{x} \in \prod_{i=1}^n \mathcal{X}_i$, we define the *social welfare* attained under $\boldsymbol{x}$ as $\mathrm{SW}(\boldsymbol{x}) := \sum_{i=1}^n u_i(\boldsymbol{x})$. The maximum possible social welfare attainable in a game $\mathcal{G}$ will be denoted by $\mathrm{OPT}_{\mathcal{G}}$. Without any essential loss of generality, it will be assumed that $\mathrm{OPT}_{\mathcal{G}} > 0$. We say that the game is *constant-sum* if $\mathrm{SW}(\boldsymbol{x}) = V \in \mathbb{R}_{>0}$ for any $\boldsymbol{x} \in \prod_{i=1}^n \mathcal{X}_i$. The *price of anarchy (PoA)* of a game $\mathcal{G}$ quantifies the loss in efficiency incurred on

account of strategic players [61]. Formally, if $\mathbb{NE}_{\mathcal{G}}$ is the nonempty set of *(mixed) Nash equilibria* of $\mathcal{G}$ (Definition 2.2) [88], we define $\mathsf{PoA}_{\mathcal{G}} := \sup_{\boldsymbol{x} \in \mathbb{NE}_{\mathcal{G}}} \left\{ \frac{\mathsf{SW}(\boldsymbol{x})}{\mathsf{OPT}_{\mathcal{G}}} \right\}$.

**Smooth games**  We are now ready to recall the seminal notion of a *smooth game*,[2] conceived in the pioneering work of Roughgarden [90] as a technique to (lower) bound the price of anarchy.

**Definition 2.1** (Smooth game [90])**.** An $n$-player game $\mathcal{G}$ is called $(\lambda, \mu)$-*smooth*, where $\lambda > 0$ and $\mu > -1$, if there exists $\boldsymbol{x}^{\star} \in \prod_{i=1}^{n} \mathcal{X}_i$ with $\mathsf{SW}(\boldsymbol{x}^{\star}) = \mathsf{OPT}_{\mathcal{G}}$ such that for every $\boldsymbol{x} \in \prod_{i=1}^{n} \mathcal{X}_i$,

$$\sum_{i=1}^{n} u_i(\boldsymbol{x}_i^{\star}, \boldsymbol{x}_{-i}) \geq \lambda \mathsf{OPT}_{\mathcal{G}} - \mu \mathsf{SW}(\boldsymbol{x}). \tag{1}$$

(In the definition above, and throughout this paper, we slightly abuse notation by parsing $u_i(\boldsymbol{x}_i^{\star}, \boldsymbol{x}_{-i})$ as $u_i(\boldsymbol{x}_1, \dots, \boldsymbol{x}_{i-1}, \boldsymbol{x}_i^{\star}, \boldsymbol{x}_{i+1}, \dots, \boldsymbol{x}_n)$.) Roughgarden [90] observed that in a $(\lambda, \mu)$-smooth game, in the sense of Definition 2.1, every Nash equilibrium attains at least a $\rho(\lambda, \mu) := \frac{\lambda}{1+\mu}$ fraction of the optimal social welfare $\mathsf{OPT}_{\mathcal{G}}$. Importantly, this efficiency guarantee immediately carries over to outcomes of no-regret learning algorithms as well. The *robust price of anarchy (rPoA$_{\mathcal{G}}$)* is the best (*i.e.*, largest) price of anarchy bound provable via a smoothness argument, and can be defined as the solution to the linear program induced by the smoothness constraints given in (1) (see (24) in Appendix A.5). One delicate point here is that the value of rPoA$_{\mathcal{G}}$ could be associated with unbounded smoothness parameters, which is a pathological and rather trivial manifestation of smoothness (Remark A.10 elaborates on this point). To be clear, when we say a game $\mathcal{G}$ attains a certain value $\rho_{\mathcal{G}}$, we mean that there exists a finite pair of legitimate smoothness parameters $(\lambda, \mu)$ such that $\rho_{\mathcal{G}} = \frac{\lambda}{1+\mu}$. With this convention, it might be the case that $\rho_{\mathcal{G}}(\lambda, \mu) \neq \mathsf{rPoA}_{\mathcal{G}}$ under any finite pair of smoothness parameters $(\lambda, \mu)$ (Remark A.10). It is also easy to see that $\mathsf{PoA}_{\mathcal{G}} \geq \mathsf{rPoA}_{\mathcal{G}}$.

**Nash equilibrium**  We next recall the concept of a *weak* Nash equilibrium, a natural generalization of the standard notion which is meaningful in multi-player games.

**Definition 2.2** (Weak Nash equilibrium [5])**.** Let $\delta \in [0, 1)$ and $\epsilon \in \mathbb{R}_{\geq 0}$. A joint strategy profile $\boldsymbol{x} \in \prod_{i=1}^{n} \mathcal{X}_i$ is an $(\epsilon, \delta)$-*weak Nash equilibrium* if at least a $1 - \delta$ fraction of the players are $\epsilon$-best responding. An $(\epsilon, 0)$-weak Nash equilibrium will be simply referred to as $\epsilon$-*Nash equilibrium*.

In the definition above, we clarify that a player $i \in [\![n]\!]$ is said to be $\epsilon$-*best responding* if $\mathsf{BRGAP}_i(\boldsymbol{x}_1, \dots, \boldsymbol{x}_n) := \max_{\boldsymbol{x}_i' \in \mathcal{X}_i} \langle \boldsymbol{x}_i', \boldsymbol{u}_i(\boldsymbol{x}_{-i}) \rangle - \langle \boldsymbol{x}_i, \boldsymbol{u}_i(\boldsymbol{x}_{-i}) \rangle \leq \epsilon$. We also define the *Nash equilibrium gap* as $\mathsf{NEGAP}(\boldsymbol{x}) := \max_{1 \leq i \leq n} \mathsf{BRGAP}_i(\boldsymbol{x})$. It has been shown that hardness results in multi-player general-sum games persist under the weak Nash equilibrium concept introduced above, even when $\epsilon$ and $\delta$ are absolute constants bounded away from 0 [4, 94].

**Regret**  We are operating in the usual online learning setting. At every time $t \in \mathbb{N}$ a player $i \in [\![n]\!]$ selects a strategy $\boldsymbol{x}_i^{(t)} \in \mathcal{X}_i$, and then receives as feedback the linear utility function $\boldsymbol{x}_i \mapsto \langle \boldsymbol{x}_i, \boldsymbol{u}_i^{(t)} \rangle$, where we overload notation so that $\boldsymbol{u}_i^{(t)} := \boldsymbol{u}_i(\boldsymbol{x}_{-i}^{(t)})$. The *regret* of player $i \in [\![n]\!]$ under a time horizon $T \in \mathbb{N}$ is defined as

$$\mathsf{Reg}_i^{(T)} := \max_{\boldsymbol{x}_i^{\star} \in \mathcal{X}_i} \left\{ \sum_{t=1}^{T} u_i(\boldsymbol{x}_i^{\star}, \boldsymbol{x}_{-i}^{(t)}) \right\} - \sum_{t=1}^{T} u_i(\boldsymbol{x}^{(t)}).$$

**Optimistic gradient descent**  By now, there are many online algorithms known to guarantee *sublinear* regret, $\mathsf{Reg}_i^{(T)} = o(T)$, even if the observed utilities are selected adversarially [18, 98]. The main no-regret learning algorithm we consider in this paper is *optimistic gradient descent* (henceforth OGD) [86, 22], which is known to yield improved regret guarantees in the setting of learning in games [103]. For each player $i \in [\![n]\!]$, OGD is defined through the following update rule for $t \in \mathbb{N}$.

$$\begin{aligned} \boldsymbol{x}_i^{(t)} &:= \mathcal{P}_{\mathcal{X}_i} \left( \hat{\boldsymbol{x}}_i^{(t)} + \eta \boldsymbol{m}_i^{(t)} \right), \\ \hat{\boldsymbol{x}}_i^{(t+1)} &:= \mathcal{P}_{\mathcal{X}_i} \left( \hat{\boldsymbol{x}}_i^{(t)} + \eta \boldsymbol{u}_i^{(t)} \right). \end{aligned} \tag{OGD}$$

---

[2] Smoothness in the sense of Definition 2.1 should not be confused with the unrelated notion of smoothness in the optimization nomenclature.

Here, $\eta > 0$ is the *learning rate*; $\boldsymbol{m}_i^{(t)} \in \mathbb{R}^{d_i}$ is the *prediction* vector; and $\hat{\boldsymbol{x}}_i^{(1)} \in \mathcal{X}_i$ is the *initialization*. It is assumed that the strategy set $\mathcal{X}_i$ is such that the Euclidean projection $\mathcal{P}_{\mathcal{X}_i}(\cdot)$ can be implemented efficiently. We will let $\boldsymbol{m}_i^{(t)} := \boldsymbol{u}_i^{(t-1)}$ for any $t \in \mathbb{N}$, where $\boldsymbol{u}_i^{(0)} := \boldsymbol{u}_i(\hat{\boldsymbol{x}}_{-i}^{(1)})$.

# 3 Convergence to Nash equilibria via smoothness

In this section, we study the convergence of optimistic gradient descent (OGD) in large games, that is to say, in the regime $n \gg 1$. In our first main result, stated below as Theorem 3.1, we show that when $\lim_{n \to +\infty} \rho_n = 1$ with a sufficiently fast rate, not only are Nash equilibria approaching the optimal social welfare, but they can also be computed efficiently in a decentralized fashion via OGD.

In the sequel, when considering a sequence of games $(\mathcal{G}_n)_{n \geq 1}$, with each game $\mathcal{G}_n$ being parameterized by the number of players $n$, we will use a subscript with variable $n$ to index the $n$th game in the sequence; that notation will also be used to refer to the other underlying parameters of the game that depend on the number of players.

**Theorem 3.1.** *Consider an $n$-player $(\lambda, \mu)$-smooth game $\mathcal{G}_n$ such that the game operator $F_n$ is $L_n$-Lipschitz continuous and $\lambda \geq (1 - \epsilon_n)(1 + \mu)$, with $\epsilon_n \neq 0$. Suppose further that all players follow OGD with learning rate $\eta_n = 1/(4L_n)$. Then, for $\delta \in (0, 1)$ and a sufficiently large number of iterations $T = O_n(nL_n^2/\gamma_n)$, where $\gamma_n := L_n\text{OPT}_{\mathcal{G}_n}\epsilon_n$, there is time $t^\star \in [\![T]\!]$ such that $\boldsymbol{x}^{(t^\star)} := (\boldsymbol{x}_1^{(t^\star)}, \ldots, \boldsymbol{x}_n^{(t^\star)})$ is a*

$$\left( \frac{1}{\sqrt{\delta}} O_n \left( \sqrt{\frac{\gamma_n}{n}} \right), \delta \right) - \text{weak Nash equilibrium.} \tag{2}$$

*In particular, if $\gamma_n = O_n(n^{1-\alpha})$, for $\alpha \in (0, 1)$, OGD yields an $(O_n(n^{-\frac{\alpha}{3}}), O_n(n^{-\frac{\alpha}{3}}))$-weak NE.*

We remark that if it further holds that $\gamma_n = o_n(1)$, the above theorem establishes convergence to the standard notion of Nash equilibrium—wherein *all* players are (almost) best responding (Corollary A.5). Parameter $\gamma_n$, which is proportional to the error term $\epsilon_n$, controls both the number of iterations and the approximation guarantee in (2); we refer to Theorem A.2 (in Appendix A) for a more precise non-asymptotic characterization, which bounds the players' cumulative best response gap as a function of the growth of $\gamma_n$. We also note that Theorem 3.1 can be strengthened so that (2) holds for *most* iterates of OGD (say 99%), not just a single one (Remark A.4).

Now, to be more concrete regarding the preconditions of Theorem 3.1, we first observe that the growth of the Lipschitz constant $L_n$ depends on the normalization assumptions as well as the structure of the underlying game $\mathcal{G}_n$. In particular, let us assume—as is standard—that the range of the utility functions is independent of $n$, in which case $\text{OPT}_{\mathcal{G}_n} = O_n(n)$. We then show that $L_n = O_n(1)$ in each of the following cases: graphical games with bounded degree (Lemma A.7), games with $O_n(1/n)$ strategic sensitivity per Definition 3.5 (Lemma A.8), and polymatrix (general-sum) games even with unbounded neighborhoods (Lemma A.9); the first two of the aforementioned classes are the most commonly studied classes in the literature under the regime $n \gg 1$. For all of those classes, applying Theorem 3.1 yields an $(o_n(1), o_n(1))$-weak Nash equilibrium for any $\epsilon_n \leq o_n(1)$. More generally, the Lipschitz constant $L_n$ can grow with $n$ (Lemma A.6), in which case $\epsilon_n$ must vanish with a faster rate for the conclusion of Theorem 3.1 to kick in. One important limitation of Theorem 3.1 is that, at least in a certain regime, it prescribes a strategy in which many players—albeit a vanishing fraction—may have a profitable deviation (in accordance with Definition 2.2); this is admittedly an inherent feature of our framework.

*Remark 3.2.* In a decentralized environment, one question that arises from Theorem 3.1 concerns the identification of a time index $t^\star \in [\![T]\!]$ that satisfies (2), which can be viewed as the stopping condition of the algorithm. We suggest two possible approaches. First, if we accept that players have access to a common source of randomness, then all players can sample the same index $t^\star$ uniformly at random from the set $[\![T]\!]$. As we point out in Remark A.4, this suffices to provide a guarantee with high probability with only a small degradation in the solution quality. The second approach, which does not rest any having a common source of randomness, involves a coordinator who can communicate with the players but possesses no information whatsoever about the underlying game. The coordinator sets a target solution quality parameterized by $(\epsilon, \delta)$ (per Definition 2.2), and after each iteration $t$ elicits from each player $i \in [\![n]\!]$ a single bit, encoding whether $\mathbb{1}\{\text{BRGAP}_i(\boldsymbol{x}^{(t)}) > \epsilon\}$. (We note that each player can indeed determine its best response gap with only its local information—namely,

the utility feedback.) The coordinator can then evaluate whether the fraction of the players with at most an $\epsilon$ best response gap matches the desired accuracy. While the second approach makes for a less decentralized protocol, the communication overhead described above is arguably very limited.

The key precondition of Theorem 3.1 pertains the behavior of the smoothness parameters, to be discussed next after we first sketch the proof of Theorem 3.1, which extends a recent technique [1].

**Proof sketch of Theorem 3.1**  The proof is based on the fact that the sum of the players' regrets $\sum_{i=1}^{n} \mathsf{Reg}_i^{(T)}$ *cannot be too negative*, which in turn follows from the assumption that $\rho_n \geq 1 - \epsilon_n$. The argument then proceeds by bounding the players' cumulative best response gap across the $T$ iterations $\sum_{t=1}^{T} \sum_{i=1}^{n} \left( \mathrm{BRGAP}_i(\boldsymbol{x}^{(t)}) \right)^2$ as a function of $\gamma_n$ and the time horizon $T \in \mathbb{N}$, ultimately leading to the conclusion of Theorem 3.1.

**Connection with the Minty property**  While we have stated Theorem 3.1 in the regime $n \gg 1$, under the premise that $\rho_n$ is sufficiently close to 1 (as a function of $n$), its conclusion is in fact interesting beyond that regime. Indeed, an important observation is that any two-player constant-sum game $\mathcal{G} := (\mathbf{A}, \mathbf{B})$, with $\langle \boldsymbol{x}_1, \mathbf{A}\boldsymbol{x}_2 \rangle + \langle \boldsymbol{x}_1, \mathbf{B}\boldsymbol{x}_2 \rangle = V$ for any $(\boldsymbol{x}_1, \boldsymbol{x}_2) \in \mathcal{X}_1 \times \mathcal{X}_2$, satisfies $\rho_{\mathcal{G}} = 1$. This is indeed a consequence of Von Neumann's minimax theorem: $\exists (\boldsymbol{x}_1^\star, \boldsymbol{x}_2^\star) \in \mathcal{X}_1 \times \mathcal{X}_2$ such that $u_1(\boldsymbol{x}_1^\star, \boldsymbol{x}_2) + u_2(\boldsymbol{x}_2^\star, \boldsymbol{x}_1) - V = \langle \boldsymbol{x}_1^\star, \mathbf{A}\boldsymbol{x}_2 \rangle - \langle \boldsymbol{x}_1, \mathbf{A}\boldsymbol{x}_2^\star \rangle \geq 0$ for any $(\boldsymbol{x}_1, \boldsymbol{x}_2) \in \mathcal{X}_1 \times \mathcal{X}_2$, in turn implying that $u_1(\boldsymbol{x}_1^\star, \boldsymbol{x}_2) + u_2(\boldsymbol{x}_2^\star, \boldsymbol{x}_1) \geq (1 + \mu)\mathrm{OPT}_{\mathcal{G}} - \mu\mathrm{SW}(\boldsymbol{x}) = V$; that is, any two-player constant-sum game is $(1 + \mu, \mu)$-smooth, thereby making the conclusion of Theorem 3.1 readily applicable (by taking $\rho \geq 1 - \epsilon^2$ for any $\epsilon > 0$; see the explicit statement of Theorem A.2). This captures and unifies earlier iteration complexity bounds under `OGD` and the extra-gradient method [15, 48, 46].

**Proposition 3.3.** *Any two-player constant-sum game $\mathcal{G}$ is $(1 + \mu, \mu)$-smooth for $\mu > -1$, implying that $\rho_{\mathcal{G}} = 1$. Thus, $O(1/\epsilon^2)$ iterations of `OGD` suffice to obtain an $\epsilon$-Nash equilibrium, for any $\epsilon > 0$.*

More broadly, there is a surprisingly overlooked but immediate connection between smooth games (per Definition 2.1) and the *Minty property*, a well-known condition in the literature on variational inequalities (VIs) [36, 66, 71]. More precisely, the Minty property postulates the existence of a strategy profile $\boldsymbol{x}^\star \in \prod_{i=1}^{n} \mathcal{X}_i$ such that $\langle \boldsymbol{x}^\star - \boldsymbol{x}, F(\boldsymbol{x}) \rangle \geq 0$ for any $\boldsymbol{x} \in \prod_{i=1}^{n} \mathcal{X}_i$, where we recall that $F$ is the operator of the game. (We caution that the last inequality is typically stated with the opposite sign since the operator $F$ is defined oppositely.) The following connection is thus immediate from the fact that $\langle \boldsymbol{x}^\star, F(\boldsymbol{x}) \rangle = \sum_{i=1}^{n} \langle \boldsymbol{x}_i^\star, \boldsymbol{u}_i(\boldsymbol{x}_{-i}) \rangle = \sum_{i=1}^{n} u_i(\boldsymbol{x}_i^\star, \boldsymbol{x}_{-i})$ and $\langle \boldsymbol{x}, F(\boldsymbol{x}) \rangle = \sum_{i=1}^{n} \langle \boldsymbol{x}_i, \boldsymbol{u}_i(\boldsymbol{x}_{-i}) \rangle = \mathrm{SW}(\boldsymbol{x})$ (by multilinearity).

**Observation 3.4.** *For any (multi-player) constant-sum game $\mathcal{G}$, the Minty property is equivalent to $\mathcal{G}$ being $(1 + \mu, \mu)$-smooth for some $\mu > -1$.*

Indeed, the Minty property implies that $\sum_{i=1}^{n} u_i(\boldsymbol{x}_i^\star, \boldsymbol{x}_{-i}) \geq V = (1 + \mu)\mathrm{OPT}_{\mathcal{G}} - \mu\mathrm{SW}(\boldsymbol{x})$, and the converse direction is also immediate. In fact, for (multi-player) zero-sum games, the Minty property is equivalent to $\mathcal{G}$ satisfying (1) under some pair $(\lambda, \mu) \in \mathbb{R}^2$. We stress that even if $\rho_{\mathcal{G}} = 1$, traditional no-regret learning algorithms such as online mirror descent do not generally enjoy iterate convergence to Nash equilibria [70], which stands in stark contrast to the behavior of `OGD` (Proposition 3.3). In light of Observation 3.4, Theorem 3.1 should also be viewed as part of an ongoing effort to establish sufficient conditions of tractability that are more permissive than the Minty property (*e.g.*, [33, 12, 14, 84, 11]). The criterion we furnish herein, based on the smoothness framework, has the important benefit of enjoying a natural economic interpretation, as well as having being extensively studied in the literature. Indeed, we will leverage insights from prior work to obtain several interesting extensions in the remainder of this section.

**Games with vanishing sensitivity**  Returning to the regime $n \gg 1$, why should we expect $\rho_n \to 1$? Indeed, if anything large games are more general than games with a small number of players since one can always incorporate "dummy" players into the game. Yet, the point is that large games oftentimes exhibit a structure that leads to more efficient outcomes. For example, one immediate implication of our framework relates to games with a *vanishing (strategic) sensitivity* (see, *e.g.*, [57, 3, 72, 31, 45]). There are various ways of defining sensitivity; here, we adopt the following standard definition.

**Definition 3.5.** The strategic *sensitivity* $\epsilon \in \mathbb{R}_{>0}$ of an $n$-player game in normal form is defined as

$$\epsilon := \max_{1 \leq i \leq n} \max_{\boldsymbol{a} \in \mathcal{A}} \max_{1 \leq i' \leq n} \max_{a'_{i'} \in \mathcal{A}_{i'}} |u_i(a'_{i'}, \boldsymbol{a}_{-i}) - u_i(\boldsymbol{a})|.$$

In words, a unilateral deviation can only impact a player's utility by an additive $\epsilon$. Now, as long as the sensitivity decays fast enough, a proof analogous to that of Theorem 3.1 implies the following.

**Theorem 3.6.** *Consider an $n$-player game $\mathcal{G}_n$ with sensitivity $\epsilon_n \in \mathbb{R}_{>0}$. Then, $T = O_n(n)$ iterations of* `OGD` *suffice to obtain a* $\left( \frac{1}{\sqrt{\delta}} O_n \left( \epsilon_n \sqrt{n} \right), \delta \right)$*-weak Nash equilibrium, for $\delta \in (0, 1)$.*

In particular, Theorem 3.6 yields an $(o_n(1), o_n(1))$-weak Nash equilibrium as long as $\epsilon_n = o_n(\sqrt{n})$. Further, in the canonical regime where $\epsilon_n = O_n(1/n)$, Theorem 3.6 circumscribes the best response gap for all but a constant number of players (by taking $\delta = O_n(1/n)$). There are many natural settings where we should expect results such as Theorem 3.6 to be applicable [57, 58, 69]. We find it conceptually compelling that our framework can provide in a unifying way equilibrium guarantees for two seemingly disparate classes of games, namely two-player zero-sum games and games with vanishing strategic sensitivity.

In a similar vein, Feldman et al. [39] showed that $\rho_n \to 1$ with a rate of $1/\sqrt{n}$ in a general auction design problem (see also [23]) under the relatively mild assumption that each player participates in the market with some constant probability (aka. probabilistic demand), thereby bypassing known barriers regarding the inefficiency of equilibria in general combinatorial domains. Their proof is based on the fact that each bidder's impact on the prices—under a simultaneous uniform-price auction format— becomes asymptotically negligible, in the spirit of Definition 3.5 introduced above. The difficulty that arises in that setting is that the natural representation of the utility functions violates our multilinearity assumption (postulated in Section 2). Instead, one would have to resort to some form of discretization before attempting to apply Theorem 3.1, which could be computationally prohibitive; the other prerequisite is that $L_n = o_n(\sqrt{n})$, for which our approach in Lemma A.8 in conjunction with the insights of Feldman et al. [39] could be useful. Understanding whether our techniques can be applied in the combinatorial auction setting of Feldman et al. [39] is left as a challenging open question.

**Efficiency of equilibria does not suffice for tractability**   A natural question arising from Theorem 3.1 is whether a similar statement applies under the assumption that $\mathsf{PoA} \to 1$, that is, assuming that all Nash equilibria of $\mathcal{G}$ are (approximately) fully efficient. This is clearly a weaker assumption, but it is unfortunately not sufficient to yield any non-trivial guarantees even in normal-form games:

**Proposition 3.7.** *Even under the promise that $\mathsf{PoA}_{\mathcal{G}} = 1$, computing a $(1/\mathrm{poly}(\mathcal{G}))$-Nash equilibrium in normal-form games in polynomial time is impossible when $n \geq 3$, unless $\mathsf{PPAD} \subseteq \mathsf{P}$.*

This stands in contrast to the class of $(1 + \mu, \mu)$-smooth games, where a fully polynomial-time approximation scheme (FPTAS) is implied by Theorem 3.1—assuming access to a utility and a projection oracle, both of which are available in, for example, most succinct normal-form games. Proposition 3.7 is a straightforward consequence of the fact that Nash equilibria are hard to compute even in constant-sum 3-player games [20]. Furthermore, in Example A.11 we identify a specific 3-player game in which `OGD` fails (unconditionally) to converge to $\epsilon$-Nash equilibria for a constant value of $\epsilon > 0$, even though $\mathsf{PoA} = 1$. Our example is based on a variant of *Shapley's game* [99].

Another notable advantage of smoothness as a criterion of convergence is that, at least when the game is represented explicitly, it is easy to compute; this is in contrast to PoA, whose identification even in two-player games is NP-hard (Proposition A.12).

**Extensions**   It turns out that smoothness per Definition 2.1 can be further sharpened using a primal-dual framework [75]. Such refined guarantees can also be translated into our setting (in the context of Theorem 3.1), as we elaborate on in Appendix A.3. The upshot is that the modification of Nadav and Roughgarden [75] necessitates analyzing the *weighted* sum of the players' regrets $\sum_{i=1}^{n} z_i \mathsf{Reg}_i^{(T)}$ under a dual set of variables $\{z_i\}_{i=1}^{n}$. Another interesting extension worth noting relies on the *local* smoothness framework [92, 78], as we explain in Appendix A.4; the key observation is that local smoothness per Nguyen [78] can be associated with a *linearized* notion of regret, at which point the analysis of Theorem 3.1 readily carries over. Finally, we also expand our scope to Bayesian mechanisms, as we expound in the upcoming subsection.

Before we proceed, it is important to point out that, unsurprisingly, smoothness is not merely enough to guarantee convergence of `OGD`; see Example A.13. It is instead crucial to additionally ensure that $\rho \approx 1$ in order to obtain interesting guarantees for the behavior of algorithms such as `OGD`.

## 3.1 Bayesian mechanisms

Next, we leverage the connection between smoothness and convergence to Nash equilibria to extend the scope of our framework to Bayesian mechanisms. In particular, analogously to Definition 2.1, Syrgkanis and Tardos [102] have introduced the notion of a *smooth mechanism* (Definition A.14); detailed background on Bayesian mechanisms and smoothness in that realm is provided in Appendix A.7. In this context, we leverage a reduction of Hartline et al. [53] from an incomplete-information to a complete-information mechanism to arrive at the following theorem. (The standard notion of a *Bayes-Nash equilibrium* is analogous to Definition 2.2, and is recalled in Definition A.15 of Appendix A.7.)

**Theorem 3.8.** *Consider a Bayesian mechanism $\mathcal{M}$ such that $\rho_{\mathcal{M}} = 1$. Then, for any $\epsilon > 0$, $T = O(1/\epsilon^2)$ iterations of* OGD *suffice to obtain an $\epsilon$-Bayes-Nash equilibrium of $\mathcal{M}$.*

In particular, OGD above is executed on the so-called *agent-form* representation of $\mathcal{M}$ (Appendix A.7). Analogously to Theorem 3.1, the above theorem can also be extended in the large $n \gg 1$ under the assumption that $\rho_n \to 1$. It is worth noting that Theorem 3.1 already can be applied to certain games of incomplete information (such as imperfect-information extensive-form games), but Theorem 3.8 additionally makes a connection with the literature on smoothness in mechanism design, which facilitates characterizing the smoothness parameters.

# 4    Improved welfare for no-regret dynamics

Roughgarden's seminal work [90] established that no-regret learning algorithms always attain asymptotically at least rPoA fraction of the optimal social welfare (on average). This guarantee is satisfactory for many classes of games where rPoA is close to 1 (emphatically those studied earlier in Section 3), but smoothness is certainly not a universal phenomenon: there are simple games in which the smoothness framework only provides vacuous guarantees; one such example is Shapley's game, discussed in Appendix A.10. As a result, one important question arising is whether it is possible to improve the efficiency bound predicted by smoothness when specific learning algorithms are in place, while at the same time still guaranteeing convergence to the set of *coarse correlated equilibria (CCE)*. We stress that optimizing over the set of CCE is typically NP-hard in succinct games [79, 6], making this question interesting also from a complexity-theoretic standpoint.

In this section, we show that it is indeed possible to obtain improved efficiency bounds under a generic condition, while at the same time guaranteeing the no-regret property for each player. A key ingredient in our improvement is the use of *clairvoyant* mirror descent, an algorithm recently introduced by Piliouras et al. [85]. More precisely, we will instantiate that algorithm with (squared) Euclidean regularization, which can be defined as follows. Let $\Pi_{\boldsymbol{x}_i}(\boldsymbol{u}_i) := \arg\max_{\boldsymbol{x}_i' \in \mathcal{X}_i} \left\{ \langle \boldsymbol{x}_i', \boldsymbol{u}_i \rangle - \frac{1}{2}\|\boldsymbol{x}_i - \boldsymbol{x}_i'\|_2^2 \right\}$ be the induced *prox operator*, where $\boldsymbol{x}_i \in \mathcal{X}_i$ and $\boldsymbol{u}_i \in \mathbb{R}^{d_i}$. *Clairvoyant gradient descent* (henceforth CGD) at time $t \in \mathbb{N}$ outputs $\boldsymbol{x}^{(t)} = \Pi_{\boldsymbol{x}^{(t-1)}}(\eta F(\boldsymbol{w}^{(t)})) := (\Pi_{\boldsymbol{x}_1^{(t-1)}}(\eta \boldsymbol{u}_1(\boldsymbol{w}_{-1}^{(t)})), \ldots, \Pi_{\boldsymbol{x}_n^{(t-1)}}(\eta \boldsymbol{u}_n(\boldsymbol{w}_{-n}^{(t)})))$, where $\boldsymbol{w}^{(t)}$ is any $\epsilon^{(t)}$-approximate fixed point of the map $\prod_{i=1}^n \mathcal{X}_i \ni \boldsymbol{w} \mapsto \Pi_{\boldsymbol{x}^{(t-1)}}(\eta F(\boldsymbol{w}))$, and $\boldsymbol{x}^{(0)} \in \mathcal{X}$ is an arbitrary initialization. It turns out that for $\eta < 1/L$, this map is a *contraction* [37, 85, 19], thereby making approximate fixed points easy to compute. Furthermore, there is also an uncoupled implementation of CGD [85], making the algorithm compelling from a decentralized standpoint as well, but we will not dwell on this issue here. We are now ready to state the main result of this section.

**Theorem 4.1.** *Suppose that all players are updating their strategies using* CGD *with $\epsilon^{(t)} \leq \frac{\min_i D_{\mathcal{X}_i}}{t^2}$ and learning rate $\eta = \frac{1}{2L}$ in a $(\lambda, \mu)$-smooth game $\mathcal{G}$, where $L$ is the Lipschtz-continuity parameter of $F$. Then, for any $\epsilon_0 > 0$ and $T \geq \frac{64L^2 D_{\mathcal{X}}^4}{\epsilon_0^2}$ iterations,*

1. *the average correlated distribution of play is a $\frac{4L D_{\mathcal{X}}^2}{T} - CCE$;*

2. *there is a time $t^\star \in \llbracket T \rrbracket$ such that*

$$\mathrm{SW}(\boldsymbol{x}^{(t^\star)}) \geq \sup_{\epsilon \geq \epsilon_0} \min \left\{ \rho_{\mathcal{G}}(\lambda, \mu) \cdot \mathrm{OPT}_{\mathcal{G}} + \frac{\epsilon^2}{16(\mu+1)L D_{\mathcal{X}}^2}, \mathsf{PoA}_{\mathcal{G}}^\epsilon \cdot \mathrm{OPT}_{\mathcal{G}} \right\}. \quad (3)$$

This is the first result that establishes simultaneously these properties under a computationally efficient algorithm, improving a recent work [1] (see also [42, 67] for related results) that failed to

guarantee convergence to CCE. In particular, that earlier work was analyzing `OGD`, and as it turns out, under a time-invariant learning rate $\eta$ it is not even known whether `OGD` ensures *sublinear* per-player regret, let alone constant (as in Corollary 4.3). The basic ingredient to this improvement is a new property of `CGD`, which we explain below. Before we sketch the proof, we note that Item 2 above can be readily strengthened so that the improvement holds for the average welfare of most of the strategies, not just a single one (Remark A.22).

**Proof sketch of Theorem 4.1**  The key step in the proof is showing (in Corollary A.21) that `CGD` satisfies the remarkable per-player regret bound $\mathrm{Reg}_i^{(T)} \leq \alpha - \gamma \sum_{t=1}^{T} \left(\mathrm{BRGAP}_i(\boldsymbol{x}^{(t)})\right)^2$, where $\alpha > 0$ depends on the approximation error of the fixed points of `CGD`—and can be made time-invariant with only an $O(\log T)$ per-iteration overhead—and $\gamma > 0$. To do this, we crucially rely on a certain property of the Euclidean regularizer (Lemma A.20), which we use in conjunction with the analysis of Farina et al. [37] who extended the original argument of Piliouras et al. [85] beyond entropic regularization.

It is worth noting that the above per-player regret bound (Corollary A.21) implies that a player with nonnegative regret will be almost always approximately best responding, a rather singular occurrence in the context of learning in games; this has interesting implications and goes well-beyond what is currently known for `OGD`. In particular, it is an open question whether Theorem 4.1 holds under `OGD`.

Next, we shall describe a concrete implication of Theorem 4.1 under a generic condition. To do so, let us denote by $\mathsf{PoA}_{\mathcal{G}}^\epsilon$ the price of anarchy in $\mathcal{G}$ with respect to the worst-case $\epsilon$-Nash equilibrium (so that $\mathsf{PoA}_{\mathcal{G}}^0 \equiv \mathsf{PoA}_{\mathcal{G}}$).

**Condition 4.2.**  *Consider a game $\mathcal{G}$ and some game-dependent parameter $C = C(\mathcal{G}) > 0$. There exists an $\epsilon_0 > 0$ such that $\mathsf{PoA}_{\mathcal{G}}^{\epsilon_0} > \mathsf{rPoA}_{\mathcal{G}} + \epsilon_0^2 C$.*

Naturally, it is always the case that $\mathsf{PoA}_{\mathcal{G}} \geq \mathsf{rPoA}_{\mathcal{G}}$. Further, $\mathsf{rPoA}_{\mathcal{G}}$ is in general strictly smaller since it measures the worst-case welfare over a larger set than $\mathsf{PoA}_{\mathcal{G}}$ (even broader than outcomes of no-regret learning); Figure 1 in Appendix A.8 further corroborates this premise in a sequence of random normal-form games. Now assuming that $\mathsf{PoA}_{\mathcal{G}} > \mathsf{rPoA}_{\mathcal{G}}$, Condition 4.2 is met if $\lim_{\epsilon \to 0} \mathsf{PoA}_{\mathcal{G}}^\epsilon = \mathsf{PoA}_{\mathcal{G}}$, a mild continuity condition (see, for example, the discussion by Roughgarden [89]).

**Corollary 4.3.**  *Consider a $(\lambda, \mu)$-smooth game $\mathcal{G}$ that satisfies Condition 4.2 under some $\epsilon_0 > 0$. Then, `CGD` after $T \geq \frac{64L^2 D_{\mathcal{X}}^4}{\epsilon_0^2}$ iterations and $\eta = \frac{1}{2L}$ satisfies the following:*

1. *the average correlated distribution of play is an $O\left(\frac{1}{T}\right)$-CCE;*
2. *there is a time $t^\star \in [\![T]\!]$ and $C'(\mathcal{G}) > 0$ such that $\mathrm{SW}(\boldsymbol{x}^{(t^\star)}) \geq (\rho_{\mathcal{G}}(\lambda, \mu) + \epsilon_0^2 C'(\mathcal{G}))\mathrm{OPT}_{\mathcal{G}}$.*

A fundamental question that arises from Theorem 4.1 is whether there exists a computationally efficient algorithm that determines a CCE with social welfare at least a PoA fraction of the optimal welfare.[3] In games where $\mathsf{PoA} = \mathsf{rPoA}$ this is clearly possible; in contrast, while Theorem 4.1 improves over the smoothness bound, it does not always guarantee welfare up to PoA. This is a central question in light of the intractability of Nash equilibria [30, 20], which has indeed served as a primary critique to the literature quantifying the price of anarchy of Nash equilibria [90].

Another promising avenue to improving the welfare predicted by the smoothness framework revolves around eliminating certain strategy profiles by arguing that they are reached with negligible probability. For example, in Appendix A.10 we identify an example where iteratively eliminating strictly dominated actions can improve the predictive power of the smoothness framework.

## 5  Further related work

**Large games**  The study of non-cooperative games with many players (*i.e.*, large games) has been a classical topic in economic theory [96, 41, 68, 95, 83, 38, 50], most recently revived in the context of *mean-field games* (*e.g.*, [64, 47, 74, 51, 16, 82, 100, 80, 29, 65]). Indeed, many traditional motivating scenarios in algorithmic game theory, including markets and Internet routing, often feature a large number of players in practice. In particular, it has emerged that, under certain conditions, equilibria

---

[3]We clarify that Theorem 4.1 could have also been stated as follows: `CGD` outputs an approximate CCE with social welfare attaining the right-hand side of (3); this is evident from the proof in Appendix A.9.

in large games exhibit certain remarkable robustness and stability properties; see, for example, the recent survey of Gradwohl and Kalai [49], as well as the older treatment of Kalai [56] on the subject. Furthermore, mechanism design in large games, along with privacy guarantees, is explored in the work of Kearns et al. [58] (see also [57, 59]).

**Efficiency in large games**   Of particular importance to our work, and specifically the precondition of Theorem 3.1, is the line of work uncovering the by now well-documented phenomenon in economics that large games exhibit, under certain relatively mild assumptions, fully efficient equilibria. Our framework additionally requires that the efficiency of equilibria can be derived via a smoothness argument, in the sense of Roughgarden [90]; we stress again that efficiency alone is of little use when it comes to equilibrium computation (Proposition 3.7). Fortunately, smoothness has emerged as the canonical paradigm for bounding the price of anarchy (*e.g.*, see the survey of Roughgarden et al. [93]), albeit with some notable exceptions [40, 55]. In particular, Feldman et al. [39] quantify the price of anarchy in large games via the smoothness framework. They show that in a general combinatorial domain with simultaneous uniform-price auctions, it holds that $\rho_n \to 1$ with a rate of $1/\sqrt{n}$ as long as there is *probabilistic demand*, meaning that every buyer abstains from the auction with a constant probability. Several other papers have studied the price of anarchy in large games [63, 23, 24, 26, 25, 17]. In particular, we highlight the work of Cole and Tao [23] which, as Feldman et al. [39], relies on a smoothness argument to establish full efficiency in the limit with a rate of $1/\sqrt{n}$ in a Walrasian auction, while asymptotic full efficiency is also shown for Fisher markets under the gross substitutes condition. Further, Carmona et al. [17] provide sufficient conditions under which equilibria are fully efficient in a class of mean-field games; understanding thus whether our framework has new implications in such games is an interesting direction for the future. We finally point out that many other papers have focused on learning in auctions and markets; see [21, 43, 104, 8, 9, 32], and references therein.

## 6   Conclusions and future work

In conclusion, we have furnished a new sufficient condition under which a family of no-regret learning algorithms, including optimistic gradient descent (OGD), approaches (weak) Nash equilibria. Our criterion has a natural economic interpretation, being intricately connected with Roughgarden's smoothness framework, and captures other well-studied conditions such as the Minty property. We have also shown that *clairvoyant* gradient descent attains an improved welfare bound compared to that predicted by the smoothness framework, while ensuring at the same time fast convergence to the set of CCE.

There are many promising directions for future work. First, we have seen that under the condition $\rho = 1$ there exists an algorithm that computes an $\epsilon$-NE in time $\mathsf{poly}(1/\epsilon)$, leading to a *pseudo* polynomial-time algorithm (under natural game representations); is there an algorithm that instead runs in time $\mathsf{poly}(\log(1/\epsilon))$?

**Convergence to Nash equilibria via computational hardness?**   Another promising approach for showing convergence to Nash equilibria is by harnessing computational hardness results for the underlying welfare maximization problem. To be specific, we consider the following condition.

**Condition 6.1.** *Consider a multi-player* $(\lambda, \mu)$*-smooth game* $\mathcal{G}$ *with* $\rho_{\mathcal{C}} \coloneqq \frac{\lambda}{1+\mu}$ *from a class of games* $\mathcal{C}$ *with the polynomial expectation property [79]. For any* $\mathcal{G} \in \mathcal{C}$*, computing a joint strategy profile* $\boldsymbol{x} \in \prod_{i=1}^{n} \mathcal{X}_i$ *such that* $\mathsf{SW}(\boldsymbol{x}) \geq \rho_{\mathcal{C}} \cdot \mathsf{OPT}_{\mathcal{G}} + 1/\mathsf{poly}(\mathcal{G})$ *is* NP*-hard, for any* $\mathsf{poly}(\mathcal{G})$*.*

Indeed, smoothness often circumscribes the welfare of polynomial algorithms, such as combinatorial auctions under XOS valuations—in fact, unconditionally under polynomial communication; see [34, Theorem 1.4] and [102, Appendix A.7]. Now, the role of Condition 6.1 is that (unless P = NP) a polynomially-bounded algorithm such as OGD—which is efficiently implementable (for games with a polynomial number of actions) under the polynomial expectation property—will have the property that $\frac{1}{T} \sum_{i=1}^{n} \mathsf{Reg}_i^{(T)} \geq -1/\mathsf{poly}(\mathcal{G})$, for any $\mathcal{G} \in \mathcal{C}$ and $\mathsf{poly}(\mathcal{G})$, which in turn leads to the following.

**Theorem 6.2.** *Consider a class* $\mathcal{C}$ *satisfying Condition 6.1. For any* $\mathcal{G} \in \mathcal{C}$ *and* $\epsilon = 1/\mathsf{poly}(\mathcal{G})$*, there is a polynomial-time algorithm for computing an* $\epsilon$*-Nash equilibrium, unless* P = NP*.*

By virtue of Corollary 4.3, the same conclusion applies even under the weaker condition that computing a CCE with welfare improving over the smoothness bound is hard; this is related to the hardness result of Barman and Ligett [6], discussed in the full version of this paper.

## Acknowledgments and Disclosure of Funding

We are grateful to anonymous reviewers and the area chair at NeurIPS for valuable feedback. We also thank Brendan Lucier for helpful pointers to the literature. This material is based on work supported by the Vannevar Bush Faculty Fellowship ONR N00014-23-1-2876, National Science Foundation grants RI-2312342 and RI-1901403, ARO award W911NF2210266, and NIH award A240108S001.

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

# A  Omitted proofs

In this section, we provide the proofs and a number of results omitted from the main body.

## A.1  Proof of Theorem 3.1

We commence with the proof of Theorem 3.1. Below, we give a more detailed version of the statement provided earlier in the main body. We first state an auxiliary lemma which will be useful for the proof, and can be extracted from earlier work [1]

**Lemma A.1.** *Suppose that player $i \in \llbracket n \rrbracket$ updates its strategy using* OGD *with learning rate $\eta > 0$. Then, for any time $t \in \mathbb{N}$,*

$$\mathrm{BRGAP}_i(\boldsymbol{x}^{(t)}) \leq \left( \frac{D_{\mathcal{X}_i}}{\eta} + \|\boldsymbol{u}_i^{(t)}\|_2 \right) \left( \|\boldsymbol{x}_i^{(t)} - \hat{\boldsymbol{x}}_i^{(t)}\|_2 + \|\boldsymbol{x}_i^{(t)} - \hat{\boldsymbol{x}}_i^{(t+1)}\|_2 \right).$$

In the sequel, we denote by $B_i \in \mathbb{R}_{>0}$ any number such that $\|\boldsymbol{u}_i(\boldsymbol{x}_{-i})\|_2 \leq B_i$, for any $\boldsymbol{x} \in \prod_{i=1}^{n} \mathcal{X}_i$. In the asymptotic notation below, we make the standard assumption that the parameters $B_i$ and $D_{\mathcal{X}_i}$ do not depend on the number of players $n$.

**Theorem A.2** (Precise version of Theorem 3.1). *Consider an $n$-player $(\lambda, \mu)$-smooth game $\mathcal{G}_n$ such that the game operator $F_n$ is $L_n$-Lipschitz continuous and $\lambda \geq (1 - \epsilon_n)(1 + \mu)$ $(\rho_n \geq 1 - \epsilon_n)$. Suppose further that all players follow* OGD *with learning rate $\eta_n = \frac{1}{4L_n}$ and any initialization $(\hat{\boldsymbol{x}}_1^{(1)}, \ldots, \hat{\boldsymbol{x}}_n^{(1)}) \in \prod_{i=1}^{n} \mathcal{X}_i$. If $\epsilon_n > 0$, then after $T := \frac{D_{\mathcal{X}}^2}{2\eta_n \epsilon_n (1+\mu)\mathrm{OPT}_{\mathcal{G}_n}}$ iterations there is a time $t^\star \in \llbracket T \rrbracket$ such that*

$$\sum_{i=1}^{n} \left( \mathrm{BRGAP}_i(\boldsymbol{x}^{(t^\star)}) \right)^2 \leq 32 \left( \frac{\max_{1 \leq i \leq n} D_{\mathcal{X}_i}^2}{(\eta_n)^2} + \max_{1 \leq i \leq n} B_i^2 \right) \eta_n \epsilon_n (1 + \mu)\mathrm{OPT}_{\mathcal{G}_n} \quad (4)$$

$$= O_n \left( L_n \mathrm{OPT}_{\mathcal{G}_n} \epsilon_n \right). \quad (5)$$

*In particular, for $\delta \in \{\frac{1}{n}, \frac{2}{n}, \ldots, 1\}$, $t^\star$ constitutes a*

$$\left( \frac{1}{\sqrt{\delta}} O_n \left( \sqrt{\frac{L_n \mathrm{OPT}_{\mathcal{G}_n} \epsilon_n}{n}} \right), \delta \right) - \text{weak Nash equilibrium.}$$

*On the other hand, if $\epsilon_n = 0$, then for any $T \in \mathbb{N}$ there is a time $t^\star \in \llbracket T \rrbracket$ such that*

$$\sum_{i=1}^{n} \left( \mathrm{BRGAP}_i(\boldsymbol{x}^{(t^\star)}) \right)^2 \leq \frac{8}{T} \left( \frac{\max_{1 \leq i \leq n} D_{\mathcal{X}_i}^2}{(\eta_n)^2} + \max_{1 \leq i \leq n} B_i^2 \right) D_{\mathcal{X}}^2.$$

Before we proceed with the proof, we note that the underlying assumption $\mu = O_n(1)$ in the asymptotic notation above is consistent with known smoothness bounds in large games [39]; we also refer to Remark A.10 for an important point regarding the range of the smoothness parameters.

*Proof of Theorem A.2.* We will first translate the assumed property $\rho_n \geq 1 - \epsilon_n$ to a lower bound on the sum of the players' regrets. In particular, we have that the sum of the players' regrets $\sum_{i=1}^{n} \mathrm{Reg}_i^{(T)}$ for any $T \in \mathbb{N}$ can be expressed as

$$\sum_{i=1}^{n} \max_{\boldsymbol{x}_i^\star \in \mathcal{X}_i} \left\{ \sum_{t=1}^{T} u_i(\boldsymbol{x}_i^\star, \boldsymbol{x}_{-i}^{(t)}) \right\} - \sum_{t=1}^{T} \mathrm{SW}(\boldsymbol{x}^{(t)}) \geq \lambda \mathrm{OPT}_{\mathcal{G}_n} T - (1 + \mu) \sum_{t=1}^{T} \mathrm{SW}(\boldsymbol{x}^{(t)}), \quad (6)$$

by Definition 2.1, where $(\lambda, \mu)$ are the assumed smoothness parameter of $\mathcal{G}_n$. Now, using the assumption that $\rho_n \geq 1 - \epsilon_n$, it follows from (6) that $\sum_{i=1}^{n} \mathrm{Reg}_i^{(T)}$ is in turn lower bounded by

$$(1 - \epsilon_n)(1 + \mu)\mathrm{OPT}_{\mathcal{G}_n} T - (1 + \mu) \sum_{t=1}^{T} \mathrm{SW}(\boldsymbol{x}^{(t)}) \geq -\epsilon_n (1 + \mu)\mathrm{OPT}_{\mathcal{G}_n} T, \quad (7)$$

where we used the fact that $\text{OPT}_{\mathcal{G}_n} \geq \text{SW}(\boldsymbol{x}^{(t)})$ (by definition of the optimal welfare). We next appropriately upper bound the sum of the players' regrets $\sum_{i=1}^{n} \text{Reg}_i^{(T)}$. Using a slight refinement of the RVU bound [103, 87], it follows that the regret $\text{Reg}_i^{(T)}$ can be upper bounded by

$$\frac{1}{2\eta_n}D_{\mathcal{X}_i}^2 + \eta_n \sum_{t=1}^{T}\|\boldsymbol{u}_i^{(t)} - \boldsymbol{m}_i^{(t)}\|_2^2 - \frac{1}{2\eta_n}\sum_{t=1}^{T}\left(\|\boldsymbol{x}_i^{(t)} - \hat{\boldsymbol{x}}_i^{(t)}\|_2^2 + \|\boldsymbol{x}_i^{(t)} - \hat{\boldsymbol{x}}_i^{(t+1)}\|_2^2\right),$$

which in turn implies that the sum of the regrets $\sum_{i=1}^{n}\text{Reg}_i^{(T)}$ is upper bounded by

$$\frac{1}{2\eta_n}\sum_{i=1}^{n}D_{\mathcal{X}_i}^2 + \eta_n\sum_{t=1}^{T}\|\boldsymbol{u}^{(t)} - \boldsymbol{m}^{(t)}\|_2^2 - \frac{1}{2\eta_n}\sum_{t=1}^{T}\left(\|\boldsymbol{x}^{(t)} - \hat{\boldsymbol{x}}^{(t)}\|_2^2 + \|\boldsymbol{x}^{(t)} - \hat{\boldsymbol{x}}^{(t+1)}\|_2^2\right), \quad (8)$$

where we have defined $\boldsymbol{u}^{(t)} \coloneqq (\boldsymbol{u}_1^{(t)}, \ldots, \boldsymbol{u}_n^{(t)})$ and $\boldsymbol{m}^{(t)} \coloneqq (\boldsymbol{m}_1^{(t)}, \ldots, \boldsymbol{m}_n^{(t)})$. Moreover, by the $L_n$-Lipschitz continuity of the game operator $F_{\mathcal{G}_n}$, it follows that $\|\boldsymbol{u}^{(t)} - \boldsymbol{m}^{(t)}\|_2 = \|F_{\mathcal{G}_n}(\boldsymbol{x}^{(t)}) - F_{\mathcal{G}_n}(\boldsymbol{x}^{(t-1)})\|_2 \leq L_n\|\boldsymbol{x}^{(t)} - \boldsymbol{x}^{(t-1)}\|_2$ for any $t \in [\![T]\!]$, where we also used the fact that $\boldsymbol{m}^{(t)} = \boldsymbol{u}^{(t-1)}$ and $\boldsymbol{x}^{(0)} \coloneqq \hat{\boldsymbol{x}}^{(1)}$. Combining with (8), it follows that for any $\eta_n \leq \frac{1}{4L_n}$,

$$\sum_{i=1}^{n}\text{Reg}_i^T \leq \frac{1}{2\eta_n}\sum_{i=1}^{n}D_{\mathcal{X}_i}^2 - \frac{1}{4\eta_n}\sum_{t=1}^{T}\left(\|\boldsymbol{x}^{(t)} - \hat{\boldsymbol{x}}^{(t)}\|_2^2 + \|\boldsymbol{x}^{(t)} - \hat{\boldsymbol{x}}^{(t+1)}\|_2^2\right). \quad (9)$$

As a result, combining (7) and (9) we have that

$$\sum_{t=1}^{T}\left(\|\boldsymbol{x}^{(t)} - \hat{\boldsymbol{x}}^{(t)}\|_2^2 + \|\boldsymbol{x}^{(t)} - \hat{\boldsymbol{x}}^{(t+1)}\|_2^2\right) \leq 2\sum_{i=1}^{n}D_{\mathcal{X}_i}^2 + 4\eta_n\epsilon_n(1+\mu)\text{OPT}_{\mathcal{G}_n}T. \quad (10)$$

Next, applying Lemma A.1 yields that

$$\sum_{t=1}^{T}\sum_{i=1}^{n}\left(\text{BRGAP}_i(\boldsymbol{x}^{(t)})\right)^2 \leq 4\left(\frac{\max_i D_{\mathcal{X}_i}^2}{\eta_n^2} + \max_i B_i^2\right)\sum_{t=1}^{T}\left(\|\boldsymbol{x}^{(t)} - \hat{\boldsymbol{x}}^{(t)}\|^2 + \|\boldsymbol{x}^{(t)} - \hat{\boldsymbol{x}}^{(t+1)}\|^2\right)$$

$$\leq 4\left(\frac{\max_i D_{\mathcal{X}_i}^2}{\eta_n^2} + \max_i B_i^2\right)\left(2D_{\mathcal{X}}^2 + 4\eta_n\epsilon_n(1+\mu)\text{OPT}_{\mathcal{G}_n}T\right), \quad (11)$$

where the first bound uses the inequality $(\alpha + \beta)^2 \leq 2\alpha^2 + 2\beta^2$, while the second bound follows from (10) by noting that $D_{\mathcal{X}}^2 = \sum_{i=1}^{n}D_{\mathcal{X}_i}^2$. As a result, we conclude that there exists $t^\star \in [\![T]\!]$, namely $t^\star \coloneqq \arg\min_{1 \leq t \leq T}\sum_{i=1}^{n}\left(\text{BRGAP}_i(\boldsymbol{x}^{(t)})\right)^2$, such that

$$\sum_{i=1}^{n}\left(\text{BRGAP}_i(\boldsymbol{x}^{(t^\star)})\right)^2 \leq 4\left(\frac{\max_{1 \leq i \leq n}D_{\mathcal{X}_i}^2}{\eta_n^2} + \max_{1 \leq i \leq n}B_i^2\right)\left(\frac{2}{T}D_{\mathcal{X}}^2 + 4\eta_n\epsilon_n(1+\mu)\text{OPT}_{\mathcal{G}_n}\right). \quad (12)$$

Now, if $\epsilon_n \neq 0$, taking $T \coloneqq \frac{D_{\mathcal{X}}^2}{2\eta_n\epsilon_n(1+\mu)\text{OPT}_{\mathcal{G}_n}} = \frac{2L_n^2 D_{\mathcal{X}}^2}{(1+\mu)\gamma_n}$ implies (4). The claimed approximation guarantee in terms of weak Nash equilibria (per Definition 2.2) in (2) can be derived from (4) as follows. We consider a parameter $\delta \in (0,1)$ so that $\delta n = \lfloor \delta n \rfloor$. Let also $\epsilon$ be the minimum among the $\delta n$ largest numbers from $\text{BRGAP}_1(\boldsymbol{x}^{(t^\star)}), \ldots, \text{BRGAP}_n(\boldsymbol{x}^{(t^\star)})$. Then, $\sum_{i=1}^{n}\left(\text{BRGAP}_i(\boldsymbol{x}^{(t^\star)})\right)^2 \geq \delta\epsilon^2 n$, and by (12) it follows that

$$\epsilon \leq \frac{1}{\sqrt{\delta}}\sqrt{32\left(\frac{\max_{1 \leq i \leq n}D_{\mathcal{X}_i}^2}{\eta_n^2} + \max_{1 \leq i \leq n}B_i^2\right)\frac{\eta_n\epsilon_n(1+\mu)\text{OPT}_{\mathcal{G}_n}}{n}}.$$

This implies (2) since $\boldsymbol{x}^{(t^\star)}$ is by definition an $(\epsilon, \delta)$-weak Nash equilibrium. Finally, if $\epsilon_n = 0$ the claimed bound follows directly from (12). $\qquad\square$

We note that the asymptotic notation in (2) and (5) applies in the regime $L_n \geq \Omega_n(1)$; this can be enforced by simply taking $L_n' \coloneqq \max\{1, L_n\}$.

*Remark* A.3. One can also introduce a variant of weak Nash equilibria which is instead parameterized by the average over the players' best response gaps $\frac{1}{n}\sum_{i=1}^{n}\mathrm{BRGAP}_i(\cdot)$. Inequality (4) implies that the average best response gap of $\boldsymbol{x}^{(t^\star)}$ can be bounded by $O_n\left(\sqrt{\frac{L_n\mathrm{OPT}_{\mathcal{G}_n}\epsilon_n}{n}}\right)$.

*Remark* A.4. While Theorem A.2 bounds the (weak) Nash equilibrium gap of a single iterate of the dynamics, (11) implies that at least a $1-\gamma$ fraction of the iterates of the dynamics constitutes a $\left(\frac{1}{\sqrt{\delta}\gamma}O_n\left(\sqrt{\frac{L_n\mathrm{OPT}_{\mathcal{G}_n}\epsilon_n}{n}}\right),\delta\right)$-weak Nash equilibrium. So, by selecting a time index $t^\star\in[\![T]\!]$ uniformly at random we obtain the desired guarantee with high probability, incurring only a small degradation in the solution quality.

In particular, if $\epsilon_n$ approaches to $0$ with a sufficiently fast rate, Theorem A.2 also implies convergence to the more standard notion of Nash equilibrium (*i.e.*, Definition 2.2 with $\delta:=0$), as we state below. In particular, the following corollary can be derived directly from (4).

**Corollary A.5.** *In the setting of Theorem A.2, if it additionally holds that $L_n\epsilon_n\mathrm{OPT}_{\mathcal{G}_n}\leq o_n(1)$, $\mathrm{OGD}$ yields an $o_n(1)$-Nash equilibrium after a sufficiently large number of iterations.*

**On the Lipschitz constant**   Next, we make some remarks regarding the dependence of the Lipschitz constant $L_n$ on the number of players in the context of general normal-form games. We first note that the Lipschitz constant $L_n$ of the underlying game operator can always be bounded as $O_n(n)$.

**Lemma A.6** (Lipschitz constant in normal-form games)**.** *For any $n$-player normal-form game $\mathcal{G}$ with utilities bounded in $[-1,1]$, the Lipschitz constant $L_n$ of the game operator satisfies $L_n\leq n\max_{1\leq i\leq n}|\mathcal{A}_i|$.*

*Proof.* For any player $i\in[\![n]\!]$ and any joint strategies $\boldsymbol{x},\boldsymbol{x}'\in\prod_{i=1}^{n}\Delta(\mathcal{A}_i)$, we have

$$\|\boldsymbol{u}_i(\boldsymbol{x}_{-i})-\boldsymbol{u}_i(\boldsymbol{x}'_{-i})\|_2\leq\sqrt{|\mathcal{A}_i|}\|\boldsymbol{u}_i(\boldsymbol{x}_{-i})-\boldsymbol{u}_i(\boldsymbol{x}'_{-i})\|_\infty \tag{13}$$

$$\leq\sqrt{|\mathcal{A}_i|}\left\|\sum_{\boldsymbol{a}_{-i}\in\mathcal{A}_{-i}}u_i(\cdot,\boldsymbol{a}_{-i})\left(\prod_{i'\neq i}\boldsymbol{x}_{i'}[a_{i'}]-\prod_{i'\neq i}\boldsymbol{x}'_{i'}[a_{i'}]\right)\right\|_1 \tag{14}$$

$$\leq\sqrt{|\mathcal{A}_i|}\left\|\sum_{\boldsymbol{a}_{-i}\in\mathcal{A}_{-i}}\left(\prod_{i'\neq i}\boldsymbol{x}_{i'}[a_{i'}]-\prod_{i'\neq i}\boldsymbol{x}'_{i'}[a_{i'}]\right)\right\|_1 \tag{15}$$

$$\leq\sqrt{|\mathcal{A}_i|}\sum_{i'\neq i}\|\boldsymbol{x}_{i'}-\boldsymbol{x}'_{i'}\|_1\leq\max_{1\leq i\leq n}|\mathcal{A}_i|\sum_{i'\neq i}\|\boldsymbol{x}_{i'}-\boldsymbol{x}'_{i'}\|_2, \tag{16}$$

where (13) uses the equivalence between the $\ell_2$ and the $\ell_\infty$ norm; (14) follows from the definition of the (expected) utility: $u_i(a_i,\boldsymbol{x}_{-i}):=\mathbb{E}_{\boldsymbol{a}_{-i}\sim\boldsymbol{x}_{-i}}[u_i(\boldsymbol{a})]=\sum_{\boldsymbol{a}_{-i}\in\mathcal{A}_{-i}}u_i(\boldsymbol{a})\prod_{i'\neq i}\boldsymbol{x}_{i'}[a_{i'}]$, for any $a_i\in\mathcal{A}_i$; (15) uses the triangle inequality along with the assumption that $|u_i(\boldsymbol{a})|\leq1$; and (16) follows from the well-known fact that the total variation distance between two product distributions can be upper bounded by the sum of the total variation distance of each individual component [54], as well as the equivalence between the $\ell_1$ and the $\ell_2$ norm. As a result, continuing from (16), we have

$$\|F(\boldsymbol{x})-F(\boldsymbol{x}')\|_2^2=\sum_{i=1}^{n}\|\boldsymbol{u}_i(\boldsymbol{x}_{-i})-\boldsymbol{u}_i(\boldsymbol{x}'_{-i})\|_2^2\leq\left(\max_{1\leq i\leq n}|\mathcal{A}_i|\right)^2\sum_{i=1}^{n}\left(\sum_{i'\neq i}\|\boldsymbol{x}_{i'}-\boldsymbol{x}'_{i'}\|_2\right)^2$$

$$\leq n^2\left(\max_{1\leq i\leq n}|\mathcal{A}_i|\right)^2\|\boldsymbol{x}-\boldsymbol{x}'\|_2^2,$$

where the last inequality used that, by Jensen's inequality, $\left(\sum_{i'\neq i}\|\boldsymbol{x}_{i'}-\boldsymbol{x}'_{i'}\|_2\right)^2\leq(n-1)\sum_{i'\neq i}\|\boldsymbol{x}_{i'}-\boldsymbol{x}'_{i'}\|_2^2\leq n\|\boldsymbol{x}-\boldsymbol{x}'\|_2^2$. This concludes the proof.  $\square$

**Graphical games**   As a byproduct of the proof above, we next point out an important refinement of Lemma A.6 concerning *graphical games*. In particular, here we assume that the utility of each player $i\in[\![n]\!]$ only depends on the actions of players belonging to its *neighborhood* $\mathcal{N}_i\subseteq[\![n]\!]\setminus\{i\}$.

We further assume that $|\mathcal{N}_i| \leq \Delta$ for any player $i \in [\![n]\!]$, where $\Delta \in \mathbb{N}$ will be referred to as the *degree* of the graphical game. To conclude the definition, we also posit that each player $i \in [\![n]\!]$ can only affect the utilities of at most $\Delta$ other players: $|i' \in [\![n]\!] : i \in \mathcal{N}_{i'}| \leq \Delta$.

**Lemma A.7** (Lipschitz constant in graphical games). *For any $n$-player graphical game with degree $\Delta \in \mathbb{N}$ and utilities bounded in $[-1, 1]$, the Lipschitz constant $L_n$ of the game operator satisfies $L_n \leq \Delta \max_{1 \leq i \leq n} |\mathcal{A}_i|$.*

*Proof.* For any player $i \in [\![n]\!]$ and any joint strategies $\boldsymbol{x}, \boldsymbol{x}' \in \prod_{i=1}^{n} \Delta(\mathcal{A}_i)$, we have

$$
\|\boldsymbol{u}_i(\boldsymbol{x}_{-i}) - \boldsymbol{u}_i(\boldsymbol{x}'_{-i})\|_2 \leq \sqrt{|\mathcal{A}_i|}\|\boldsymbol{u}_i(\boldsymbol{x}_{-i}) - \boldsymbol{u}_i(\boldsymbol{x}'_{-i})\|_\infty
$$

$$
\leq \sqrt{|\mathcal{A}_i|} \left\| \sum_{\prod_{i' \in \mathcal{N}_i} \mathcal{A}_{i'}} \left( \prod_{i' \in \mathcal{N}_i} \boldsymbol{x}_{i'}[a_{i'}] - \prod_{i' \in \mathcal{N}_i} \boldsymbol{x}'_{i'}[a_{i'}] \right) \right\|_1
$$

$$
\leq \sqrt{|\mathcal{A}_i|} \sum_{i' \in \mathcal{N}_i} \|\boldsymbol{x}_{i'} - \boldsymbol{x}'_{i'}\|_1 \leq \max_{1 \leq i \leq n} |\mathcal{A}_i| \sum_{i' \in \mathcal{N}_i} \|\boldsymbol{x}_{i'} - \boldsymbol{x}'_{i'}\|_2,
$$

where the derivation above is similar to that in Lemma A.6. As a result,

$$
\|F(\boldsymbol{x}) - F(\boldsymbol{x}')\|_2^2 = \sum_{i=1}^{n} \|\boldsymbol{u}_i(\boldsymbol{x}_{-i}) - \boldsymbol{u}_i(\boldsymbol{x}'_{-i})\|_2^2
$$

$$
\leq \left( \max_{1 \leq i \leq n} |\mathcal{A}_i| \right)^2 \sum_{i=1}^{n} \left( \sum_{i' \in \mathcal{N}_i} \|\boldsymbol{x}_{i'} - \boldsymbol{x}'_{i'}\|_2 \right)^2
$$

$$
\leq \Delta \left( \max_{1 \leq i \leq n} |\mathcal{A}_i| \right)^2 \sum_{i=1}^{n} \sum_{i' \in \mathcal{N}_i} \|\boldsymbol{x}_{i'} - \boldsymbol{x}'_{i'}\|_2^2
$$

$$
= \Delta \left( \max_{1 \leq i \leq n} |\mathcal{A}_i| \right)^2 \sum_{i=1}^{n} \|\boldsymbol{x}_i - \boldsymbol{x}'_i\|_2^2 |i' \in [\![n]\!] : i \in \mathcal{N}_{i'}|
$$

$$
\leq \Delta^2 \left( \max_{1 \leq i \leq n} |\mathcal{A}_i| \right)^2 \|\boldsymbol{x} - \boldsymbol{x}'\|_2^2.
$$

$\square$

**Games with vanishing sensitivity** We next focus on a different subclass of normal-form games; namely, games with small sensitivity (per Definition 3.5). Taking a step back, in graphical games every player can only be impacted by (and have an impact to) a small number of other players. Instead, here we consider games where a player's utility can be impacted by all other players, but only by a small amount.

**Lemma A.8** (Lipschitz constant in games with vanishing sensitivity). *For any $n$-player normal-form game with sensitivity $\epsilon_n \in \mathbb{R}_{>0}$, the Lipschitz constant $L_n$ of the game operator satisfies $L_n \leq \epsilon_n n \max_{1 \leq i \leq n} |\mathcal{A}_i|$.*

*Proof.* Let $i \in [\![n]\!]$. For $\boldsymbol{x}_1, \boldsymbol{x}'_1 \in \Delta(\mathcal{A}_1)$ and $\boldsymbol{a}_{-1} \in \mathcal{A}_{-1}$ (restricting on Player 1 here is without any loss, and only made for the sake of simplicity in the notation), it follows that

$$
u_i(\boldsymbol{x}_1, \boldsymbol{a}_{-1}) - u_i(\boldsymbol{x}'_1, \boldsymbol{a}_{-1}) = \sum_{a_1 \in \mathcal{A}_1} \boldsymbol{x}_1[a_1] u_i(a_1, \boldsymbol{a}_{-1}) - \sum_{a_1 \in \mathcal{A}_1} \boldsymbol{x}'_1[a_1] u_i(a_1, \boldsymbol{a}_{-1})
$$

$$
= \sum_{a_1 \in \mathcal{A}_1 \setminus \{a'_1\}} (\boldsymbol{x}_1[a_1] - \boldsymbol{x}'_1[a_1]) u_i(a_1, \cdot) + (\boldsymbol{x}_1[a'_1] - \boldsymbol{x}'_1[a'_1]) u_i(a'_1, \cdot)
$$

$$
= \sum_{a_1 \in \mathcal{A}_1 \setminus \{a'_1\}} (\boldsymbol{x}_1[a_1] - \boldsymbol{x}'_1[a_1])(u_i(a_1, \boldsymbol{a}_{-1}) - u_i(a'_1, \boldsymbol{a}_{-1})), \quad (17)
$$

for some $a_1' \in \mathcal{A}_1$, where we used that $(\boldsymbol{x}_1[a_1'] - \boldsymbol{x}_1'[a_1']) = \sum_{a_1 \in \mathcal{A}_1 \setminus \{a_1'\}} (\boldsymbol{x}_1'[a_1] - \boldsymbol{x}_1[a_1])$ since $\boldsymbol{x}_1, \boldsymbol{x}_1' \in \Delta(\mathcal{A}_1)$. Continuing from (17), we have

$$
\begin{aligned}
|u_i(\boldsymbol{x}_1, \boldsymbol{a}_{-1}) - u_i(\boldsymbol{x}_1', \boldsymbol{a}_{-1})| &\leq \sum_{a_1 \in \mathcal{A}_1 \setminus \{a_1'\}} |\boldsymbol{x}_1[a_1] - \boldsymbol{x}_1'[a_1]| |u_i(a_1, \boldsymbol{a}_{-1}) - u_i(a_1', \boldsymbol{a}_{-1})| \\
&\leq \epsilon_n \sum_{a_1 \in \mathcal{A}_1 \setminus \{a_1'\}} |\boldsymbol{x}_1[a_1] - \boldsymbol{x}_1'[a_1]| \leq \epsilon_n \|\boldsymbol{x}_1 - \boldsymbol{x}_1'\|_1,
\end{aligned}
$$

where $\epsilon_n$ is the sensitivity of the game. Similar reasoning yields that $|u_i(\boldsymbol{x}) - u_i(\boldsymbol{x}_1', \boldsymbol{x}_{-1})| \leq \epsilon_n \|\boldsymbol{x}_1 - \boldsymbol{x}_1'\|_1$, for any $\boldsymbol{x}_{-1} \in \prod_{i=2}^n \Delta(\mathcal{A}_i)$. As a result, we have

$$
\begin{aligned}
\|\boldsymbol{u}_1(\boldsymbol{x}_{-1}) - \boldsymbol{u}_1(\boldsymbol{x}_{-1}')\|_\infty &\leq |u_1(\cdot, \boldsymbol{x}_2, \boldsymbol{x}_3, \ldots, \boldsymbol{x}_n) - u_1(\cdot, \boldsymbol{x}_2', \boldsymbol{x}_3, \ldots, \boldsymbol{x}_n)| \\
&\quad + |u_1(\cdot, \boldsymbol{x}_2', \boldsymbol{x}_3, \ldots, \boldsymbol{x}_n) - u_1(\cdot, \boldsymbol{x}_2', \boldsymbol{x}_3', \ldots, \boldsymbol{x}_n)| \\
&\quad + \ldots \\
&\quad + |u_1(\cdot, \boldsymbol{x}_2', \boldsymbol{x}_3', \ldots, \boldsymbol{x}_{n-1}' \boldsymbol{x}_n) - u_1(\cdot, \boldsymbol{x}_2', \boldsymbol{x}_3', \ldots, \boldsymbol{x}_n')| \\
&\leq \epsilon_n \sum_{i \neq 1} \|\boldsymbol{x}_i - \boldsymbol{x}_i'\|_1.
\end{aligned}
$$

By symmetry, we have shown that $\|\boldsymbol{u}_i(\boldsymbol{x}_{-i}) - \boldsymbol{u}_i(\boldsymbol{x}_{-i}')\|_\infty \leq \epsilon_n \sum_{i' \neq i} \|\boldsymbol{x}_{i'} - \boldsymbol{x}_{i'}'\|_1$, and the rest of the argument is identical to that of Lemma A.6. $\qquad \square$

**Polymatrix games** A careful examination of the proof of Lemma A.8 reveals that its conclusion in fact applies under a more relaxed condition compared to what imposed by Definition 3.5; namely, we can define

$$
\epsilon := \max_{1 \leq i \leq n} \max_{\boldsymbol{a} \in \mathcal{A}} \max_{i' \neq i} \max_{a_{i'}' \in \mathcal{A}_{i'}} |u_i(\boldsymbol{a}) - u_i(a_{i'}', \boldsymbol{a}_{-i'})|. \tag{18}
$$

In words, when considering the utility of a player $i \in [\![n]\!]$, we only bound deviations by players besides $i$. It is easy to see that Lemma A.8 in fact applies even if $\epsilon \in \mathbb{R}_{>0}$ is defined as in (18). This observation enables capturing other interesting classes of games under the premise that $L_n = O_n(1)$, such as *polymatrix games*. Specifically, a polymatrix game is defined with respect to an underlying directed graph $G = ([\![n]\!], E)$, so that each node of $G$ is uniquely associated with the corresponding player. For every edge $(i, i') \in E$ there is a matrix $\mathbf{A}_{i,i'} \in \mathbb{R}^{\mathcal{A}_i \times \mathcal{A}_{i'}}$ so that the utility of Player $i$ is defined as

$$
u_i(\boldsymbol{x}) := \frac{1}{n} \sum_{i' \in \mathcal{N}_i} \boldsymbol{x}_i^\top \mathbf{A}_{i,i'} \boldsymbol{x}_{i'}, \tag{19}
$$

where $\mathcal{N}_i := \{i' \in [\![n]\!] : (i, i') \in E\}$. Unlike the class of graphical games we saw earlier, here we do not restrict the size of the neighborhoods. For this reason, we have normalized each player's utility by a $1/n$ factor in (19), for otherwise the utilities are not guaranteed to be bounded (independent of the number of players $n$). It is then easy to see that polymatrix games are subject to Lemma A.8 under $\epsilon$ defined in (18), which here satisfies $\epsilon = O(1/n)$. Below, we provide a simpler and sharper argument compared to Lemma A.8.

**Lemma A.9.** *For any $n$-player polymatrix game, the Lipschitz constant $L_n$ of the game operator satisfies $L_n \leq \max_{(i,i') \in E} \|\mathbf{A}_{i,i'}\|_2$, where $\| \cdot \|_2$ here denotes the spectral norm.*

*Proof.* By definition of the utility functions in (19), we have that for any player $i \in [\![n]\!]$ and $\boldsymbol{x}, \boldsymbol{x}' \in \prod_{i=1}^n \Delta(\mathcal{A}_i)$,

$$
\begin{aligned}
\|\boldsymbol{u}_i(\boldsymbol{x}_{-i}) - \boldsymbol{u}_i(\boldsymbol{x}'_{-i})\|_2 &\leq \frac{1}{n} \left\| \sum_{i' \in \mathcal{N}_i} \mathbf{A}_{i,i'}(\boldsymbol{x}_{i'} - \boldsymbol{x}'_{i'}) \right\|_2 \\
&\leq \frac{1}{n} \sum_{i' \in \mathcal{N}_i} \|\mathbf{A}_{i,i'}(\boldsymbol{x}_{i'} - \boldsymbol{x}'_{i'})\|_2 \\
&\leq \frac{1}{n} \sum_{i' \in \mathcal{N}_i} \|\mathbf{A}_{i,i'}\|_2 \|\boldsymbol{x}_{i'} - \boldsymbol{x}'_{i'}\|_2 \\
&\leq \frac{\max_{(i,i') \in E} \|\mathbf{A}_{i,i'}\|_2}{n} \sum_{i' \in \mathcal{N}_i} \|\boldsymbol{x}_{i'} - \boldsymbol{x}'_{i'}\|_2,
\end{aligned}
$$

and the claim follows. $\qquad\square$

## A.2   Games with vanishing sensitivity

We next establish an important implication of our framework concerning the class of games with vanishing strategic sensitivity (per Definition 3.5); the statement of the theorem is recalled below.

**Theorem 3.6.** *Consider an $n$-player game $\mathcal{G}_n$ with sensitivity $\epsilon_n \in \mathbb{R}_{>0}$. Then, $T = O_n(n)$ iterations of* OGD *suffice to obtain a* $\left(\frac{1}{\sqrt{\delta}} O_n\left(\epsilon_n \sqrt{n}\right), \delta\right)$*-weak Nash equilibrium, for $\delta \in (0, 1)$.*

*Proof.* Let $i \in [\![n]\!]$. Following the proof of Lemma A.8, the definition of sensitivity implies that $|u_i(\boldsymbol{x}'_i, \boldsymbol{x}_{-i}) - u_i(\boldsymbol{x})| \leq \epsilon_n \|\boldsymbol{x}'_i - \boldsymbol{x}_i\|_1 \leq 2\epsilon_n$, for any $\boldsymbol{x} \in \prod_{i=1}^n \Delta(\mathcal{A}_i)$ and $\boldsymbol{x}'_i \in \Delta(\mathcal{A}_i)$. The proof now follows that of Theorem A.2. In particular, we can lower bound the sum of the players' regrets as follows.

$$
\begin{aligned}
\sum_{i=1}^n \mathsf{Reg}_i^{(T)} &= \sum_{i=1}^n \left( \max_{\boldsymbol{x}_i^\star \in \Delta(\mathcal{A}_i)} \left\{ \sum_{t=1}^T u_i(\boldsymbol{x}_i^\star, \boldsymbol{x}_{-i}^{(t)}) \right\} - \sum_{t=1}^T u_i(\boldsymbol{x}^{(t)}) \right) \\
&\geq - \sum_{i=1}^n \sum_{t=1}^T |u_i(\boldsymbol{x}'_i, \boldsymbol{x}_{-i}^{(t)}) - u_i(\boldsymbol{x}^{(t)})| \geq -2Tn\epsilon_n.
\end{aligned}
$$

As a result, following the proof of Theorem A.2, we conclude that for learning rate $\eta_n := \frac{1}{4L_n}$ there exists a time $t^\star \in [\![T]\!]$ such that

$$
\sum_{i=1}^n \left( \mathrm{BRGAP}_i(\boldsymbol{x}^{(t^\star)}) \right)^2 \leq 4 \left( \frac{\max_{1 \leq i \leq n} D_{\mathcal{X}_i}^2}{\eta_n^2} + \max_{1 \leq i \leq n} B_i^2 \right) \left( \frac{2D_{\mathcal{X}}^2}{T} + 4\eta_n n \epsilon_n \right),
$$

Thus, setting

$$
T := \frac{D_{\mathcal{X}}^2}{2\eta_n n \epsilon_n} = \frac{2D_{\mathcal{X}}^2 L_n}{n\epsilon_n} \leq 2D_{\mathcal{X}}^2 \left( \max_{1 \leq i \leq n} |\mathcal{A}_i| \right) = O_n(n),
$$

by Lemma A.8, yields that

$$
\sum_{i=1}^n \left( \mathrm{BRGAP}_i(\boldsymbol{x}^{(t^\star)}) \right)^2 \leq O_n(L_n n \epsilon_n) = O_n(n^2 \epsilon_n^2),
$$

where the asymptotic notation above applies in the regime $L_n \geq \Omega_n(1)$. The statement thus follows. $\qquad\square$

We note that the conclusion of Theorem 3.6 also applies under the weaker notion of sensitivity defined in (18), if we additionally assume that there exists $\boldsymbol{x}^\star \in \prod_{i=1}^n \Delta(\mathcal{A}_i)$ such that for any $\boldsymbol{x} \in \prod_{i=1}^n \Delta(\mathcal{A}_i)$,

$$
\sum_{i=1}^n u_i(\boldsymbol{x}_i^\star, \boldsymbol{x}_{-i}) - \sum_{i=1}^n u_i(\boldsymbol{x}) \geq -n\epsilon_n. \tag{20}
$$

Condition (20) can be met even in games with large sensitivity (Observation 3.4), which is why we chose to state Theorem 3.6 under a stronger but more interpretable condition.

Next, we point out two interesting extensions of our approach based on developments following the original smoothness framework of Roughgarden [90].

### A.3 Refined smoothness

Throughout this paper we have relied on the original smoothness framework [90] to derive much of our results. Nevertheless, it is worth noting that there are certain extensions documented in the literature that can make our predictions sharper. To be more precise, focusing on normal-form games, Nadav and Roughgarden [75] noted that the original smoothness framework is in fact able to provide efficiency bounds for a set of equilibria even more permissive than CCE, which they refer to as *average* CCE with respect to a welfare-maximizing strategy $\boldsymbol{x}^\star \in \prod_{i=1}^n \Delta(\mathcal{A}_i)$ (ACCE*). In particular, the latter relaxation only requires a guarantee for the average regret over the players, $\frac{1}{n} \sum_{i=1}^n \mathsf{Reg}_i^{(T)}(\boldsymbol{x}_i^\star)$, while CCE instead requires a guarantee for the maximum regret $\max_{1 \leq i \leq n} \mathsf{Reg}_i^{(T)}$. Based on this observation, Nadav and Roughgarden [75] developed a primal-dual framework in order to provide refined guarantees (beyond the original smoothness framework) for CCE*, which boils down to solving the following linear program.

$$
\begin{aligned}
\text{maximize} \quad & \rho \\
\text{subject to} \quad & \textstyle\sum_{i=1}^n z_i \left( u_i(a_i^\star, \boldsymbol{a}_{-i}) - u_i(\boldsymbol{a}) \right) \geq \rho \mathrm{SW}(\boldsymbol{a}^\star) - \mathrm{SW}(\boldsymbol{a}), \boldsymbol{a} \in \textstyle\prod_{i=1}^n \mathcal{A}_i, \\
& z_i \geq 0.
\end{aligned}
\tag{21}
$$

Here, it is assumed that the underlying game is in normal form, with $\prod_{i=1}^n \mathcal{A}_i$ being the set of joint action profiles, and $\boldsymbol{a}^\star \in \prod_{i=1}^n \mathcal{A}_i$ a welfare-maximizing joint action. (The formulation of Nadav and Roughgarden [75] is based on cost-minimization games, but of course their LP can be cast directly for payoff-maximization games in the form of (21).) The above LP is a simple transformation of the fractional-linear program corresponding to optimizing $\rho := \frac{\lambda}{1+\mu}$ subject to the smoothness constraints of Definition 2.1, but with the additional flexibility of optimizing over a vector $\boldsymbol{z} \in \mathbb{R}^n_{\geq 0}$. In particular, when restricting $z_1 = z_2 = \cdots = z_n$, this exactly recovers the LP for computing the robust price of anarchy—given in (24). Nevertheless, Nadav and Roughgarden [75] pointed out that the additional flexibility of (21) can have an arbitrarily large impact on the predicted efficiency [75, Remark 1].

The sharper definition of smoothness introduced by Nadav and Roughgarden [75] can be leveraged in our framework as follows. We can define a generalized robust price of anarchy as the solution of the LP (21). If this quantity approaches to 1 with a sufficiently fast rate (in terms of the number of players), we can extend Theorem A.2 by analyzing the *weighted* sum of the players' regrets $\sum_{i=1}^n z_i \mathsf{Reg}_i^{(T)}$. It is easy to see that the argument of Theorem 3.1 readily carries over under the condition that the ratio $z_i/z_{i'}$ is bounded for any $i, i' \in [\![n]\!]$, as well as the assumption that each $z_i$ is bounded away from 0. In contrast, if there exists a pair $i, i' \in [\![n]\!]$ for which the ratio $z_i/z_{i'}$ is unbounded, our current techniques do not appear to be applicable; this is related to the well-known difficulty of deriving so-called RVU bounds for the maximum of the players' regret, instead of their sum [103]. In other words, our current techniques can sharpen Theorem 3.1 by considering solutions of (21) under the additional (linear) constraint that the variables $\{z_i\}_{i=1}^n$ have bounded pairwise ratio. Given that the more general framework of Nadav and Roughgarden [75] leads to improved bounds, we expect that this modification should have applications in our setting as well. In particular, we point out that the 2-player example of Nadav and Roughgarden [75] that separates ACCE* from CCE* indeed satisfies $z_1/z_2 \approx 2.3$ [75, Proposition 2], so the separation manifests itself even when the pairwise ratio is bounded by an absolute constant.

We finally refer to the works of Nguyen [78] and Kulkarni and Mirrokni [62] for a different primal-dual take on smoothness.

### A.4 Local smoothness

Another interesting extension of our techniques can be obtained using the framework of *local smoothness*, first introduced by Roughgarden and Schoppmann [92] and recently refined by Nguyen [78] in the context of splittable congestion games. In what follows, we follow the treatment of Nguyen

[78] as it fits our framework. In this context, Nguyen [78] introduced the notion of a $(\lambda, \mu)$-*dual-smooth* differentiable utility function $u : \mathbb{R}_{\geq 0} \to \mathbb{R}_{\geq 0}$ with the property that for every vectors $\boldsymbol{z} = (z_1, \ldots, z_n) \in \mathbb{R}_{\geq 0}^n$ and $\boldsymbol{z}' = (z_1', \ldots, z_n') \in \mathbb{R}_{\geq 0}^n$,

$$z'u(z) + \sum_{i=1}^{n} z_i(z_i' - z_i)u'(z) \geq \lambda z'u(z') - \mu z u(z), \tag{22}$$

where $z := \sum_{i=1}^{n} z_i$ and $z' := \sum_{i=1}^{n} z_i'$. We also note that $u'$ above denotes the derivative of $u$. (Again, (22) has been translated to utility-maximization games compared to the original formulation of Nguyen [78].) Furthermore, a splittable congestion game is called $(\lambda, \mu)$-dual-smooth [78] if for every resource $e \in E$ the utility function $u_e : \mathbb{R}_{\geq 0} \to \mathbb{R}_{\geq 0}$ is $(\lambda, \mu)$-dual-smooth in the sense of (22). The importance of Nguyen's extension is that it can be applied to *coarse* correlated equilibria, as opposed to the original definition of Roughgarden and Schoppmann [92] that was applicable to correlated equilibria; this distinction is incidentally important for our framework.

We will connect this concept of local smoothness with the linearization of the players' regrets. In particular, the utility of a player $i \in [\![n]\!]$ under a joint strategy $\boldsymbol{x}$ is defined as $u_i(\boldsymbol{x}) := \sum_{e \in E} \boldsymbol{x}_i[e] u_e(\sum_{i=1}^{n} \boldsymbol{x}_i[e])$. Then,

$$\frac{\partial u_i(\boldsymbol{x})}{\partial \boldsymbol{x}_i[e]} = u_e\left(\sum_{i=1}^{n} \boldsymbol{x}_i[e]\right) + \boldsymbol{x}_i[e] u_e'\left(\sum_{i=1}^{n} \boldsymbol{x}_i[e]\right).$$

As a result, for any $t \in \mathbb{N}$,

$$\sum_{i=1}^{n} \langle \boldsymbol{x}_i^\star - \boldsymbol{x}_i^{(t)}, \nabla_{\boldsymbol{x}_i} u_i(\boldsymbol{x}^{(t)}) \rangle = \sum_{e \in E} u_e(\boldsymbol{x}[e])\boldsymbol{x}^\star[e] - \sum_{e \in E} u_e(\boldsymbol{x}[e])\boldsymbol{x}[e]$$

$$+ \sum_{e \in E} \sum_{i=1}^{n} \boldsymbol{x}_i[e](\boldsymbol{x}_i^\star[e] - \boldsymbol{x}_i[e])u_e'(\boldsymbol{x}[e]),$$

with the understanding that $\boldsymbol{x}[e] := \sum_{i=1}^{n} \boldsymbol{x}_i[e]$ and $\boldsymbol{x}^\star[e] := \sum_{i=1}^{n} \boldsymbol{x}_i^\star[e]$ for any $e \in E$. Now let us define $\mathsf{Reg}_{\mathcal{L},i}^{(T)}(\boldsymbol{x}_i^\star) := \sum_{t=1}^{T} \langle \boldsymbol{x}_i^\star - \boldsymbol{x}_i^{(t)}, \nabla_{\boldsymbol{x}_i} u_i(\boldsymbol{x}) \rangle$. Combining the last displayed equality with local smoothness (22), we get that

$$\sum_{i=1}^{n} \mathsf{Reg}_{\mathcal{L},i}^{(T)}(\boldsymbol{x}_i^\star) \geq \lambda \sum_{e \in E} \boldsymbol{x}^\star[e] u_e(\boldsymbol{x}^\star[e]) - (\mu + 1) \sum_{e \in E} \boldsymbol{x}[e] u_e(\boldsymbol{x}[e])$$

$$= \lambda \mathrm{SW}(\boldsymbol{x}^\star) - (\mu + 1)\mathrm{SW}(\boldsymbol{x}). \tag{23}$$

Consequently, if we define a local smoothness bound, $\rho_{\mathcal{L}} := \frac{\lambda}{1+\mu}$ with $(\lambda, \mu)$ being subject to (22), Theorem 3.1 can be readily extended in the regime $\rho_{\mathcal{L}} \to 1$ based on (23).

### A.5 Considerations based on PoA

It is natural to ask if the conclusion of Theorem 3.1 can be relaxed to PoA $\to 1$. Here, we point out that such an assumption does not suffice to obtain interesting guarantees. In particular, we next note Proposition 3.7, stated earlier in the main body, which is an immediate byproduct of the hardness of computing Nash equilibria in constant-sum games.

**Proposition 3.7.** *Even under the promise that* $\mathsf{PoA}_{\mathcal{G}} = 1$, *computing a* $(1/\mathrm{poly}(\mathcal{G}))$-*Nash equilibrium in normal-form games in polynomial time is impossible when* $n \geq 3$, *unless* $\mathsf{PPAD} \subseteq \mathsf{P}$.

*Proof.* Chen et al. [20] showed that computing a $(1/\mathrm{poly}(\max_i |\mathcal{A}_i|))$-Nash equilibrium in a general-sum two-player game in normal form is PPAD-hard. As a result, the same applies in constant-sum 3-player games by suitably incorporating an additional player who has no strategic impact on the game. Further, in any constant-sum game $\mathcal{G}$ it clearly holds that $\mathsf{PoA}_{\mathcal{G}} = 1$, concluding the proof. $\square$

*Remark* A.10. The positive result established in Theorem A.2 requires that $\rho \approx 1$ under a bounded pair of smoothness parameters $(\lambda, \mu)$. One may wonder whether similar conclusions apply even

when the smoothness parameters are unbounded. In particular, the *robust price of anarchy (*rPoA*)* can be defined as the solution to the following linear program.

$$
\begin{aligned}
\text{maximize} \quad & \rho \\
\text{subject to} \quad & z \sum_{i=1}^{n} \left( u_i(a_i^\star, \boldsymbol{a}_{-i}) - u_i(\boldsymbol{a}) \right) \geq \rho \text{SW}(\boldsymbol{a}^\star) - \text{SW}(\boldsymbol{a}), \boldsymbol{a} \in \textstyle\prod_{i=1}^{n} \mathcal{A}_i, \\
& z \geq 0.
\end{aligned}
\tag{24}
$$

Above $\boldsymbol{a}^\star = (a_1^\star, \ldots, a_n^\star) \in \prod_{i=1}^{n} \mathcal{A}_i$ is a welfare-maximizing action profile, which can be assumed to be unique for the purposes of our discussion here. In this context, can we extend the conclusion of Theorem A.2 under the assumption that $\text{rPoA}_{\mathcal{G}_n} \to 1$? In general, that is not possible. Indeed, in any constant-sum game one can make the LP (24) feasible by taking $z = 0$ and $\rho = 1$; that is, $\text{rPoA}_{\mathcal{G}_n} = 1$ for any constant-sum game $\mathcal{G}$. Thus, assuming merely that $\text{rPoA}_{\mathcal{G}} \to 1$ is not enough to obtain interesting guarantees for computing Nash equilibria (in accordance with Proposition 3.7). In other words, our underlying assumption that the smoothness parameters are bounded is necessary. To further explain this discrepancy, we note that a constant-sum game is generally not $(1 + \mu, \mu)$-smooth (for a finite $\mu > -1$), for otherwise any constant-sum game would satisfy the Minty property (Proposition 3.3), which would in turn contradict Proposition 3.7. The case where $z = 0$ in any optimal solution of (24) essentially corresponds to a pathological manifestation of smoothness in which the underlying parameters are unbounded. We stress again that throughout this paper, when we say that a game is $(\lambda, \mu)$-smooth we, of course, posit that those smoothness parameters are finite. Further, when we say that $\rho_{\mathcal{G}} = 1$ in a game $\mathcal{G}$, we accept that there are finite smoothness parameters associated with $\rho_{\mathcal{G}}$. With this convention, we reiterate that there are games in which $\text{rPoA}_{\mathcal{G}} \neq \rho_{\mathcal{G}}$.

Next, we provide a concrete example based on Shapley's game [99] in which OGD (unconditionally) fails to converge to an $\epsilon$-Nash equilibrium, for a constant $\epsilon > 0$.

*Example* A.11. We consider a 3-player game $\mathcal{G}$ defined as follows. We let

$$
\mathbf{A} := \begin{bmatrix} 1 & 1 & 2 \\ 2 & 1 & 1 \\ 1 & 2 & 1 \end{bmatrix}, \mathbf{B} := \begin{bmatrix} 1 & 2 & 1 \\ 1 & 1 & 2 \\ 2 & 1 & 1 \end{bmatrix}.
\tag{25}
$$

Then, we set $u_1(\boldsymbol{x}_1, \boldsymbol{x}_2, \boldsymbol{x}_3) := \boldsymbol{x}_1^\top \mathbf{A} \boldsymbol{x}_2$, $u_2(\boldsymbol{x}_1, \boldsymbol{x}_2, \boldsymbol{x}_3) := \boldsymbol{x}_1^\top \mathbf{B} \boldsymbol{x}_2$, and $u_3(\boldsymbol{x}_1, \boldsymbol{x}_2, \boldsymbol{x}_3) := 3 - \boldsymbol{x}_1^\top \mathbf{A} \boldsymbol{x}_2 - \boldsymbol{x}_1^\top \mathbf{B} \boldsymbol{x}_2$. Thus, for any joint strategy $(\boldsymbol{x}_1, \boldsymbol{x}_2, \boldsymbol{x}_3)$ it holds that $\text{SW}(\boldsymbol{x}_1, \boldsymbol{x}_2, \boldsymbol{x}_3) = 3$, implying that $\text{PoA}_{\mathcal{G}} = 1$. Further, $\mathcal{G}$ is in normal form. Now, through a numerical simulation we draw the following conclusion: Although $\text{PoA}_{\mathcal{G}} = 1$, for $T \gg 1$ OGD with learning rate $\eta := 0.01$ and initialization $(\hat{\boldsymbol{x}}_1^{(1)}, \hat{\boldsymbol{x}}_2^{(1)}, \cdot) := ((0.5, 0.25, 0.25), (0.25, 0.5, 0.25), \cdot)$ satisfies $\text{NEGAP}(\boldsymbol{x}^{(t)}) \geq 0.1875$ for any $t \in [\![T]\!]$, where $(\boldsymbol{x}^{(t)})_{t \geq 1}$ is the sequence of iterates produced by OGD. We note that here we do not consider the initialization from the uniform distribution simply because that happens to be the unique Nash equilibrium in Shapley's game; the conclusion above readily applies for any initialization by suitably modifying the underlying game. We also note that the specific value for the learning rate specified above is used for concreteness, and the conclusion is not tied to that specific value.

We next note another separation between the bound predicted by smoothness and PoA, which gives an additional reason why the smoothness framework is more suited as a criterion for determining tractability. Below, we make for simplicity the assumption that the game has a unique welfare-maximizing action profile, which can always be enforced by incorporating an arbitrarily small noise in the players' utilities.

**Proposition A.12.** *Determining whether a game is $(\lambda, \mu)$-smooth can be done in polynomial time in explicitly represented normal-form games. In contrast, even in two-player games, determining* PoA *is* NP-*hard.*

*Proof.* First of all, the welfare-maximizing action profile $\boldsymbol{a}^\star \in \prod_{i=1}^{n} \mathcal{A}_i$ can be trivially computed in polynomial time (in the size of the input) since the game is explicitly represented. For a legitimate $(\lambda, \mu)$ pair, determining whether the game is $(\lambda, \mu)$-smooth can be phrased as a feasibility linear program, with a number of constraints equal to the number of possible joint action profiles, each corresponding to a separate constraint in (1); namely, $\sum_{i=1}^{n} u_i(a_i^\star, \boldsymbol{a}_{-i}) \geq \lambda \text{OPT} - \mu \sum_{i=1}^{n} u_i(\boldsymbol{a})$. As a result, the number of constraints is polynomial in the description of the game (since it is assumed that the game is explicitly represented). Furthermore, one can optimize over the smoothness parameters by considering the LP (of polynomial size) given in (24), which determines the robust price

of anarchy (rPoA). In accordance with Theorem A.2 (see Remark A.10), one can also incorporate the constraint $z \geq 1/\mathrm{poly}(\mathcal{G})$. Regarding the second claim, hardness of determining PoA even in two-player games follows directly from the reduction of Conitzer and Sandholm [27, Theorem 1]. In particular, an algorithm computing PoA would enable determining the satisfiability of a SAT formula. □

One important question is whether smoothness can be identified in polynomial time even in *succinctly* represented games [79]. Indeed, the obvious algorithm for identifying smoothness described above requires a number of constraints that scales exponentially with the number of players, which is especially problematic in the regime of large games we focus on in Section 3.

We need to clarify, however, that knowing the smoothness parameters is certainly not a prerequisite for applying our approach. In detail, one can first of all apply Theorem 3.1 in classes of games where there are available analytical bounds for $\rho_n$ as a function of $n$, obviating the need to determine whether $\rho_n$ is close to $1$. Besides this point, and more importantly, one can always execute the algorithm prescribed by Theorem 3.1—which does not require any knowledge regarding the smoothness parameters—and then efficiently evaluate the solution quality (per Definition 2.2). If the desired accuracy has been reached, then this is precisely the initial goal; otherwise, we can safely assume that the preconditons of Theorem 3.1 are not met.

### A.6 Smoothness does not suffice for convergence

Next, we provide for completeness an example of a smooth game where OGD fails to converge to Nash equilibria. This shows that, as expected, it is not merely enough to know that $\rho \neq 0$ to obtain interesting guarantees.

*Example* A.13. This example is based on a bimatrix game in normal form described with the payoff matrices

$$\mathbf{A} = \begin{bmatrix} 0.2 & 0.8 & 0.9 & 0.3 \\ 0.2 & 0.8 & 0.2 & 0.3 \\ 0.9 & 0.2 & 0.4 & 0.4 \\ 0.6 & 0.9 & 0.3 & 0.1 \end{bmatrix}, \mathbf{B} = \begin{bmatrix} 0.4 & 0.2 & 0 & 0.1 \\ 0.5 & 0 & 0.2 & 0.8 \\ 0.7 & 0.8 & 0 & 0.4 \\ 0 & 0 & 0.1 & 0.4 \end{bmatrix}. \tag{26}$$

We first claim that this bimatrix game $\mathcal{G}$ satisfies $\rho_{\mathcal{G}} \geq 0.125$. Indeed, we first see that the welfare-maximizing profile of (26) reads $(\boldsymbol{x}_1^\star, \boldsymbol{x}_2^\star) = ((0, 0, 1, 0), (1, 0, 0, 0))$. We also claim that for any pair of actions $a_1 \in \mathcal{A}_1$ and $a_2 \in \mathcal{A}_2$ it holds that

$$\sum_{i=1}^{2} u_i(\boldsymbol{x}_i^\star, a_{-i}) \geq 0.125 \cdot \mathrm{OPT}_{\mathcal{G}},$$

where $\mathrm{OPT}_{\mathcal{G}} = 1.6$. As a result, it follows that for any $\boldsymbol{x}_1 \in \Delta(\mathcal{A}_1)$ and $\boldsymbol{x}_2 \in \Delta(\mathcal{A}_2)$ it holds that $\mathbb{E}_{a_1 \sim \boldsymbol{x}_1, a_2 \sim \boldsymbol{x}_2}[\sum_{i=1}^{2} u_i(\boldsymbol{x}_i^\star, a_{-i})] \geq 0.125 \cdot \mathrm{OPT}_{\mathcal{G}}$, in turn implying that $\sum_{i=1}^{2} u_i(\boldsymbol{x}_i^\star, \boldsymbol{x}_{-i}) \geq 0.125 \cdot \mathrm{OPT}_{\mathcal{G}}$. This means that $\mathcal{G}$ is $(0.125, 0)$-smooth. Furthermore, through a numerical simulation we draw the following conclusion: for $T \gg 1$, OGD with learning rate $\eta := 0.01$ satisfies $\mathrm{NEGAP}(\boldsymbol{x}_1^{(t)}, \boldsymbol{x}_2^{(t)}) \geq 0.046$ for any $t \in [\![T]\!]$, where $(\boldsymbol{x}_1^{(t)}, \boldsymbol{x}_2^{(t)})_{t \geq 1}$ is the sequence of iterates produced by OGD. Once again, it is worth noting that this conclusion is not tied to the specific choice of learning rate we chose above, which is only used for concreteness.

### A.7 Bayesian mechanisms

In this subsection, we consider the standard *independent private value* model of Bayesian mechanisms. In particular, each player $i \in [\![n]\!]$ has a *type* $v_i$ drawn from a distribution $\mathcal{F}_i$ over a finite set of types $\mathcal{V}_i$; without any loss, we may assume that $\mathcal{F}_i$ is the uniform distribution over $\mathcal{V}_i$. It is further assumed that players' types are pairwise independent. After each player $i \in [\![n]\!]$ draws a type $v_i \sim \mathcal{F}_i$, $i$ selects an action $a_i(v_i) \in \mathcal{A}_i$, which is a function of its type $v_i$. Now consider a fixed profile of types $\boldsymbol{v} \in \mathcal{V} := \mathcal{V}_1 \times \mathcal{V}_2 \ldots \mathcal{V}_n$. The (expected) utility of Player $i$ under a joint strategy profile $\boldsymbol{x}(\boldsymbol{v})$ is denoted by $u_i(\boldsymbol{x}; v_i) := \mathbb{E}_{\boldsymbol{a} \sim \boldsymbol{x}}[u_i(\boldsymbol{a}; v_i)]$. There is also a *principal* agent who does not take an action in the game, and whose utility under $\boldsymbol{x}$ is given by $R(\boldsymbol{x}) := \mathbb{E}_{\boldsymbol{a} \sim \boldsymbol{x}}[R(\boldsymbol{a})]$. Accordingly, the social welfare is defined as $\mathrm{SW}(\boldsymbol{x}, \boldsymbol{v}) := \sum_{i=1}^{n} u_i(\boldsymbol{x}; v_i) + R(\boldsymbol{x})$, while $\mathrm{OPT}_{\mathcal{M}}(\boldsymbol{v})$ represents the optimal social welfare of mechanism $\mathcal{M}$ as a function of the joint type $\boldsymbol{v} \in \mathcal{V}$. We will make the assumption that the utility functions assign solely nonnegative values.

**Smooth mechanisms**    Analogously to the notion of a smooth game (Definition 2.1), Syrgkanis and Tardos [102] introduced the notion of a smooth mechanism, formally recalled below.

**Definition A.14** (Smooth mechanism [102])**.** A mechanism $\mathcal{M}$ is $(\lambda, \mu)$-smooth, where $\lambda, \mu \geq 0$, if there exists an strategy profile $\boldsymbol{x}^\star(\boldsymbol{v}) \in \prod_{i=1}^n \Delta(\mathcal{A}_i)$, for every type profile $\boldsymbol{v} \in \mathcal{V}$, such that for any action profile $\boldsymbol{a} \in \mathcal{A}$ and type profile $\boldsymbol{v} = (v_1, \dots, v_n) \in \mathcal{V}$,

$$\sum_{i=1}^n u_i(\boldsymbol{x}_i^\star(\boldsymbol{v}), \boldsymbol{a}_{-i}; v_i) \geq \lambda \mathrm{OPT}_{\mathcal{M}}(\boldsymbol{v}) - \mu R(\boldsymbol{a}). \tag{27}$$

As it turns out, many important mechanisms satisfy the above definition under various parameters $(\lambda, \mu)$ [102]. For a $(\lambda, \mu)$-smooth mechanism $\mathcal{M}$, we define $\rho_{\mathcal{M}} := \frac{\lambda}{\max\{1, \mu\}}$.

**Definition A.15.** Let $\boldsymbol{x}_i : \mathcal{V}_i \to \Delta(\mathcal{A}_i)$ be the strategy of each player $i \in [\![n]\!]$. A joint strategy profile $\boldsymbol{x}$ is a *Bayes-Nash equilibrium (BNE)* if for any player $i \in [\![n]\!]$, any type $v_i \in \mathcal{V}_i$ and deviation $a_i' \in \mathcal{A}_i$,

$$\mathbb{E}_{\boldsymbol{v}_{-i} \sim \mathcal{F}_{-i}}[u_i(\boldsymbol{x}(\boldsymbol{v}); v_i)] \geq \mathbb{E}_{\boldsymbol{v}_{-i} \sim \mathcal{F}_{-i}}[u_i(a_i', \boldsymbol{x}_{-i}(\boldsymbol{v}_{-i}); v_i)]. \tag{28}$$

This definition coincides with the standard notion of Nash equilibrium we saw in Definition 2.2 when each distribution $\mathcal{F}_i$ is a point mass. We also note that an $\epsilon$-BNE incorporates an $\epsilon \geq 0$ additive slackness in (28).

**Population interpretation of Bayesian games**    In the *agent-form* representation of a Bayesian game [53], it is assumed that there are $n$ finite subpopulations of players, each corresponding to a player $i \in [\![n]\!]$. Each player belonging to population $i$ corresponds to a type $v_i \in \mathcal{V}_i$, which is distinct in each population and across populations. In this induced population game, nature first draws one player from each population, and then each player $v_i$ selects an action $a_i(v_i)$; the game is then played under those selected actions. In symbols, the utility of Player $v_i$ from population $i$ reads

$$u_{i, v_i}^{\mathsf{AG}}(\boldsymbol{a}) := \mathbb{E}_{\boldsymbol{v} \sim \mathcal{F}}[u_i(\boldsymbol{a}(\boldsymbol{v}); v_i) \mathbb{1}\{v_i\}], \tag{29}$$

where by $\{v_i\}$ above we denote the event that type $v_i$ is selected by nature among the population corresponding to Player $i$. The importance of the population interpretation of the Bayesian game is that it induces a mechanism of complete information, which will be denoted by $\mathcal{M}^{\mathsf{AG}} = \mathcal{M}^{\mathsf{AG}}(\mathcal{M})$. The following characterization highlights an important connection between $\mathcal{M}^{\mathsf{AG}}$ and $\mathcal{M}$.

**Theorem A.16** ([53])**.** *If a mechanism $\mathcal{M}$ is $(\lambda, \mu)$-smooth, then the complete information mechanism $\mathcal{M}^{\mathsf{AG}} = \mathcal{M}^{\mathsf{AG}}(\mathcal{M})$ is also $(\lambda, \mu)$-smooth.*

We now proceed with the proof of Theorem 3.8. To keep the exposition self-contained, we will not make explicit use of Theorem A.16.

**Theorem 3.8.** *Consider a Bayesian mechanism $\mathcal{M}$ such that $\rho_{\mathcal{M}} = 1$. Then, for any $\epsilon > 0$, $T = O(1/\epsilon^2)$ iterations of* `OGD` *suffice to obtain an $\epsilon$-Bayes-Nash equilibrium of $\mathcal{M}$.*

*Proof.* Under the assumption that $\rho_{\mathcal{M}} = 1$, it follows that there exists a pair $(\lambda, \mu) \in \mathbb{R}_{\geq 0}^2$ such that $\mathcal{M}$ is $(\lambda, \mu)$-smooth with $\lambda = \max\{1, \mu\}$. Further, by definition it holds that $\mathrm{OPT}_{\mathcal{M}}(\boldsymbol{v}) \geq \sum_{i=1}^n u_i(\boldsymbol{a}; v_i) + R(\boldsymbol{a})$, for any action profile $\boldsymbol{a} \in \mathcal{A}$. Combining with (27), we have that there exists a strategy profile $\boldsymbol{x}^\star(\boldsymbol{v}) \in \prod_{i=1}^n \Delta(\mathcal{A}_i)$ such that for every type profile $\boldsymbol{v} \in \mathcal{V}$ and action profile $\boldsymbol{a} \in \mathcal{A}$,

$$\sum_{i=1}^n u_i(\boldsymbol{x}_i^\star(\boldsymbol{v}), \boldsymbol{a}_{-i}; v_i) \geq \lambda \sum_{i=1}^n u_i(\boldsymbol{a}; v_i) + \lambda R(\boldsymbol{a}) - \mu R(\boldsymbol{a}) \geq \sum_{i=1}^n u_i(\boldsymbol{a}; v_i),$$

since $\lambda = \max\{1, \mu\}$ and utilities are nonnegative. As a result, for any $\boldsymbol{x}(\boldsymbol{v}) \in \prod_{i=1}^n \Delta(\mathcal{A}_i)$,

$$\sum_{i=1}^n \mathbb{E}_{\boldsymbol{v} \sim \mathcal{F}}[u_i(\boldsymbol{x}_i^\star(\boldsymbol{v}), \boldsymbol{x}_{-i}(\boldsymbol{v}_{-i}); v_i)] \geq \sum_{i=1}^n \mathbb{E}_{\boldsymbol{v} \sim \mathcal{F}}[u_i(\boldsymbol{x}(\boldsymbol{v}); v_i)]. \tag{30}$$

Further, by definition of the agent-form utilities (29) and the law of total expectation, (30) can be equivalently cast as

$$\sum_{i=1}^n \sum_{v_i \in \mathcal{V}_i} u_{i, v_i}^{\mathsf{AG}}(\boldsymbol{x}_{i, v_i}^\star, \boldsymbol{x}_{-(i, v_i)}) \geq \sum_{i=1}^n \sum_{v_i \in \mathcal{V}_i} u_{i, v_i}^{\mathsf{AG}}(\boldsymbol{x}),$$

where we used the notation $\boldsymbol{x}_{i,v_i} \coloneqq \boldsymbol{x}_i(v_i)$. This implies that the sum of the players' regrets in the agent-form representation is nonnegative. The proof of Theorem 3.8 then follows from the correspondence between Nash equilibria in the agent-form representation of $\mathcal{M}$ and Bayes-Nash equilibria in the original incomplete-information mechanism $\mathcal{M}$. $\square$

## A.8 PoA vs robust PoA

In this subsection, we provide some additional justification for the condition rPoA $\neq$ PoA required in Corollary 4.3. In particular, we conduct experiments on a set of random normal-form games. Some illustrative results for 10 random games are demonstrated in Figure 1. Overall, we observe that not only rPoA $\neq$ PoA, but in fact the gap between the two quantities is typically substantial. This discrepancy is expected since, as we have already stressed, rPoA quantifies the worst-case welfare over a (typically) much broader set of equilibria.

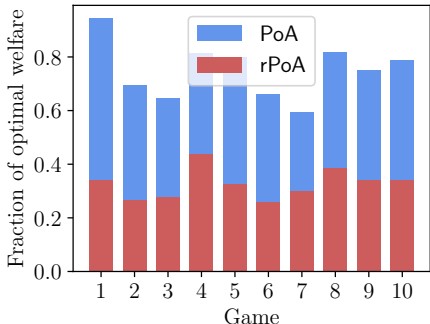

Figure 1: PoA versus rPoA in random normal-form games.

## A.9 Proof of Theorem 4.1

In this subsection, we provide the proof of Theorem 4.1. For completeness, we also include the proofs of certain results known from prior work [37, 85, 77].

To this end, we first recall that the *prox* operator associated with the (squared) Euclidean regularizer $\frac{1}{2}\|\cdot\|_2^2$ is defined as

$$\Pi_{\boldsymbol{x}_i}(\boldsymbol{u}_i) \coloneqq \arg\max_{\boldsymbol{x}_i' \in \mathcal{X}_i} \left\{ \langle \boldsymbol{x}_i', \boldsymbol{u}_i \rangle - \frac{1}{2}\|\boldsymbol{x}_i - \boldsymbol{x}_i'\|_2^2 \right\}, \tag{31}$$

under some utility vector $\boldsymbol{u}_i \in \mathbb{R}^{d_i}$. Accordingly, we let $\Pi_{\boldsymbol{x}}(\boldsymbol{u}) \coloneqq (\Pi_{\boldsymbol{x}_1}(\boldsymbol{u}_1), \ldots, \Pi_{\boldsymbol{x}_n}(\boldsymbol{u}_n))$, where $\boldsymbol{u} \coloneqq (\boldsymbol{u}_1, \ldots, \boldsymbol{u}_n)$. With this definition in mind, we note the following property of the prox operator.

**Lemma A.17** ([37, 77]). *The prox operator $\Pi_{\boldsymbol{x}}(\cdot)$ is 1-Lipschitz continuous with respect to $\|\cdot\|_2$ for any $\boldsymbol{x} \in \prod_{i=1}^n \mathcal{X}_i$.*

*Proof.* Let $i \in [\![n]\!]$. For any $\boldsymbol{u}_i, \boldsymbol{u}_i' \in \mathbb{R}^{d_i}$ and $\boldsymbol{x}_i \in \mathcal{X}_i$,

$$\|\Pi_{\boldsymbol{x}_i}(\boldsymbol{u}_i) - \Pi_{\boldsymbol{x}_i}(\boldsymbol{u}_i')\|_2 = \|\mathcal{P}_{\mathcal{X}_i}(\boldsymbol{x}_i + \boldsymbol{u}_i) - \mathcal{P}_{\mathcal{X}_i}(\boldsymbol{x}_i + \boldsymbol{u}_i')\|_2 \leq \|\boldsymbol{u}_i - \boldsymbol{u}_i'\|_2,$$

where we used the fact that the projection operator is non-expansive with respect to $\|\cdot\|_2$, and that

$$\begin{aligned}
\Pi_{\boldsymbol{x}_i}(\boldsymbol{u}_i) = \arg\max_{\boldsymbol{x}_i' \in \mathcal{X}_i} \left\{ \langle \boldsymbol{x}_i', \boldsymbol{u}_i \rangle - \frac{1}{2}\|\boldsymbol{x}_i - \boldsymbol{x}_i'\|_2^2 \right\} &= \arg\max_{\boldsymbol{x}_i' \in \mathcal{X}_i} \left\{ -\frac{1}{2}\|\boldsymbol{x}_i'\|_2^2 + \langle \boldsymbol{x}_i', \boldsymbol{u}_i + \boldsymbol{x}_i \rangle \right\} \\
&= \arg\min_{\boldsymbol{x}_i' \in \mathcal{X}_i} \left\{ \|\boldsymbol{x}_i' - (\boldsymbol{u}_i + \boldsymbol{x}_i)\|_2^2 \right\} \\
&= \mathcal{P}_{\mathcal{X}_i}(\boldsymbol{x}_i + \boldsymbol{u}_i).
\end{aligned}$$

This implies that $\|\Pi_{\boldsymbol{x}}(\boldsymbol{u}) - \Pi_{\boldsymbol{x}}(\boldsymbol{u}')\|_2 \leq \|\boldsymbol{u} - \boldsymbol{u}'\|$, where $\boldsymbol{u} = (\boldsymbol{u}_1, \ldots, \boldsymbol{u}_n)$ and $\boldsymbol{u}' = (\boldsymbol{u}_1', \ldots, \boldsymbol{u}_n')$. $\square$

Using this lemma, a key observation for the analysis of *clairvoyant* mirror descent is the following contraction property [85, 37].

**Proposition A.18** ([85, 37]). *Suppose that $F$ is $L$-Lipschitz continuous. For any $\boldsymbol{x}' \in \prod_{i=1}^{n} \mathcal{X}_i$ the function*

$$\prod_{i=1}^{n} \mathcal{X}_i \ni \boldsymbol{w} \mapsto \Pi_{\boldsymbol{x}'}(\eta F(\boldsymbol{w})) \tag{32}$$

*is $(\eta L)$-Lipschitz continuous. As a result, function (32) is a contraction mapping as long as $\eta < \frac{1}{L}$.*

This follows directly from Lemma A.17: $\|\Pi_{\boldsymbol{x}'}(\eta F(\boldsymbol{w})) - \Pi_{\boldsymbol{x}'}(\eta F(\boldsymbol{w}'))\|_2 \leq \eta \|F(\boldsymbol{w}) - F(\boldsymbol{w}')\|_2 \leq \eta L \|\boldsymbol{w} - \boldsymbol{w}'\|_2$.

Proposition A.18 reassures us that fixed points of the contraction mapping (32) not only exist, but $\epsilon$-approximate fixed points can also be computed in a time proportional to $\log(1/\epsilon)$ [85]. In this context, if $\boldsymbol{x}^{(0)} \in \prod_{i=1}^{n} \mathcal{X}_i$ is an arbitrary point, we consider the update rule defined for $t \in \mathbb{N}$ via

$$\boldsymbol{x}^{(t)} = \Pi_{\boldsymbol{x}^{(t-1)}}(\eta F(\boldsymbol{w}^{(t)})), \tag{33}$$

where $\boldsymbol{w}^{(t)} \in \prod_{i=1}^{n} \mathcal{X}_i$ is any point such that $\|\boldsymbol{w}^{(t)} - \Pi_{\boldsymbol{x}^{(t-1)}}(\eta F(\boldsymbol{w}^{(t)}))\|_2 \leq \epsilon^{(t)}$. It is important to note that this sequence $(\boldsymbol{x}^{(t)})_{t \geq 1}$ is not uniquely defined, but with a slight abuse we refer to any sequence satisfying (33) as *clairvoyant gradient descent (*CGD*)*. CGD satisfies the following remarkable regret bound.

**Theorem A.19** ([85, 37]). *For any $T \in \mathbb{N}$, the regret of player $i \in \llbracket n \rrbracket$ under CGD satisfies*

$$\mathsf{Reg}_i^{(T)} \leq \frac{D_{\mathcal{X}_i}^2}{2\eta} - \frac{1}{2\eta} \sum_{t=1}^{T} \|\boldsymbol{x}_i^{(t)} - \boldsymbol{x}_i^{(t-1)}\|_2^2 + D_{\mathcal{X}_i} L_i \sum_{t=1}^{T} \epsilon^{(t)}, \tag{34}$$

*where $L_i \in \mathbb{R}_{>0}$ is the Lipschitz constant of $\boldsymbol{u}_i$ with respect to $\|\cdot\|_2$.*

*Proof.* Given that $\boldsymbol{x}_i^{(t)} := \Pi_{\boldsymbol{x}_i^{(t-1)}}(\eta \boldsymbol{u}_i(\boldsymbol{w}_{-i}^{(t)}))$, the first-order optimality condition implies that for any $\boldsymbol{x}_i^{\star} \in \mathcal{X}_i$ and $t \in \llbracket T \rrbracket$,

$$\left\langle \boldsymbol{x}_i^{(t)} - \boldsymbol{x}_i^{\star}, \eta \boldsymbol{u}_i(\boldsymbol{w}_{-i}^{(t)}) - (\boldsymbol{x}_i^{(t)} - \boldsymbol{x}_i^{(t-1)}) \right\rangle \geq 0. \tag{35}$$

We further have that

$$\langle \boldsymbol{x}_i^{\star} - \boldsymbol{x}_i^{(t)}, \boldsymbol{x}_i^{(t)} - \boldsymbol{x}_i^{(t-1)} \rangle = -\frac{1}{2} \|\boldsymbol{x}_i^{\star} - \boldsymbol{x}_i^{(t)}\|_2^2 + \frac{1}{2} \|\boldsymbol{x}_i^{\star} - \boldsymbol{x}_i^{(t-1)}\|_2^2 - \frac{1}{2} \|\boldsymbol{x}_i^{(t)} - \boldsymbol{x}_i^{(t-1)}\|_2^2,$$

thereby implying through a telescopic summation that

$$\sum_{t=1}^{T} \langle \boldsymbol{x}_i^{\star} - \boldsymbol{x}_i^{(t)}, \boldsymbol{x}_i^{(t)} - \boldsymbol{x}_i^{(t-1)} \rangle \leq \frac{1}{2} D_{\mathcal{X}_i}^2 - \frac{1}{2} \sum_{t=1}^{T} \|\boldsymbol{x}_i^{(t)} - \boldsymbol{x}_i^{(t-1)}\|_2^2.$$

Combining with (35),

$$\sum_{t=1}^{T} \langle \boldsymbol{x}_i^{\star} - \boldsymbol{x}_i^{(t)}, \boldsymbol{u}_i(\boldsymbol{w}_{-i}^{(t)}) \rangle \leq \frac{1}{2\eta} D_{\mathcal{X}_i}^2 - \frac{1}{2\eta} \sum_{t=1}^{T} \|\boldsymbol{x}_i^{(t)} - \boldsymbol{x}_i^{(t-1)}\|_2^2.$$

The claim thus follows from the fact that

$$\sum_{t=1}^{T} \langle \boldsymbol{x}_i^{\star} - \boldsymbol{x}_i^{(t)}, \boldsymbol{u}_i(\boldsymbol{w}_{-i}^{(t)}) - \boldsymbol{u}_i(\boldsymbol{x}_{-i}^{(t)}) \rangle \geq -\sum_{t=1}^{T} \|\boldsymbol{x}_i^{\star} - \boldsymbol{x}_i^{(t)}\|_2 \|\boldsymbol{u}_i(\boldsymbol{w}_{-i}^{(t)}) - \boldsymbol{u}_i(\boldsymbol{x}_{-i}^{(t)})\|_2$$

$$\geq -D_{\mathcal{X}_i} L_i \sum_{t=1}^{T} \epsilon^{(t)}.$$

$\square$

Unlike prior work, here we will make crucial use of the negative term in (34). The important lemma below will enable us to express the regret bound (34) in terms of $i$'s best response gap; it is analogous to Lemma A.1 we stated earlier for OGD.

**Lemma A.20.** *Fix any $t \in \mathbb{N}$ and let $\boldsymbol{x}^{(t)} = \Pi_{\boldsymbol{x}^{(t-1)}}(\eta F(\boldsymbol{w}^{(t)}))$, where $\boldsymbol{w}^{(t)} \in \prod_{i=1}^{n} \mathcal{X}_i$ is any point such that $\|\boldsymbol{w}^{(t)} - \Pi_{\boldsymbol{x}^{(t-1)}}(\eta F(\boldsymbol{w}^{(t)}))\|_2 \leq \epsilon^{(t)}$. Then,*

$$\mathrm{BRGAP}_i(\boldsymbol{x}^{(t)}) \leq D_{\mathcal{X}_i} \|\boldsymbol{u}_i(\boldsymbol{w}_{-i}^{(t)}) - \boldsymbol{u}_i(\boldsymbol{x}_{-i}^{(t)})\|_2 + \frac{D_{\mathcal{X}_i}}{\eta}\|\boldsymbol{x}_i^{(t)} - \boldsymbol{x}_i^{(t-1)}\|_2.$$

*Proof.* By the first-order optimality condition, it follows that for any $\boldsymbol{x}_i^{\star} \in \mathcal{X}_i$,

$$\left\langle \boldsymbol{x}_i^{(t)} - \boldsymbol{x}_i^{\star}, \eta \boldsymbol{u}_i(\boldsymbol{w}_{-i}^{(t)}) - (\boldsymbol{x}_i^{(t)} - \boldsymbol{x}_i^{(t-1)}) \right\rangle \geq 0,$$

in turn implying that

$$\langle \boldsymbol{x}_i^{(t)} - \boldsymbol{x}_i^{\star}, \boldsymbol{u}_i(\boldsymbol{w}_{-i}^{(t)}) \rangle \geq -\frac{1}{\eta}\langle \boldsymbol{x}_i^{\star} - \boldsymbol{x}_i^{(t)}, \boldsymbol{x}_i^{(t)} - \boldsymbol{x}_i^{(t-1)} \rangle. \tag{36}$$

Furthermore,

$$\begin{aligned}
\langle \boldsymbol{x}_i^{(t)} - \boldsymbol{x}_i^{\star}, \boldsymbol{u}_i(\boldsymbol{w}_{-i}^{(t)}) \rangle &= \langle \boldsymbol{x}_i^{(t)} - \boldsymbol{x}_i^{\star}, \boldsymbol{u}_i(\boldsymbol{x}_{-i}^{(t)}) \rangle + \langle \boldsymbol{x}_i^{(t)} - \boldsymbol{x}_i^{\star}, \boldsymbol{u}_i(\boldsymbol{w}_{-i}^{(t)}) - \boldsymbol{u}_i(\boldsymbol{x}_{-i}^{(t)}) \rangle \\
&\leq \langle \boldsymbol{x}_i^{(t)} - \boldsymbol{x}_i^{\star}, \boldsymbol{u}_i(\boldsymbol{x}_{-i}^{(t)}) \rangle + \|\boldsymbol{x}_i^{(t)} - \boldsymbol{x}_i^{\star}\|_2 \|\boldsymbol{u}_i(\boldsymbol{w}_{-i}^{(t)}) - \boldsymbol{u}_i(\boldsymbol{x}_{-i}^{(t)})\|_2 \\
&\leq \langle \boldsymbol{x}_i^{(t)} - \boldsymbol{x}_i^{\star}, \boldsymbol{u}_i(\boldsymbol{x}_{-i}^{(t)}) \rangle + D_{\mathcal{X}_i} \|\boldsymbol{u}_i(\boldsymbol{w}_{-i}^{(t)}) - \boldsymbol{u}_i(\boldsymbol{x}_{-i}^{(t)})\|_2. \tag{37}
\end{aligned}$$

Combining (36) and (37) yields that

$$\langle \boldsymbol{x}_i^{(t)}, \boldsymbol{u}_i(\boldsymbol{x}_{-i}^{(t)}) \rangle - \max_{\boldsymbol{x}_i^{\star} \in \mathcal{X}_i} \langle \boldsymbol{x}_i^{\star}, \boldsymbol{u}_i(\boldsymbol{x}_{-i}^{(t)}) \rangle \geq -D_{\mathcal{X}_i} \|\boldsymbol{u}_i(\boldsymbol{w}_{-i}^{(t)}) - \boldsymbol{u}_i(\boldsymbol{x}_{-i}^{(t)})\|_2 - \frac{D_{\mathcal{X}_i}}{\eta}\|\boldsymbol{x}_i^{(t)} - \boldsymbol{x}_i^{(t-1)}\|_2,$$

concluding the proof. $\qquad\square$

**Corollary A.21.** *For any $T \in \mathbb{N}$, the regret of player $i \in \llbracket n \rrbracket$ under CGD can be bounded as*

$$\mathrm{Reg}_i^{(T)} \leq \frac{D_{\mathcal{X}_i}^2}{2\eta} - \frac{\eta}{4D_{\mathcal{X}_i}^2} \sum_{t=1}^{T} \left( \mathrm{BRGAP}_i(\boldsymbol{x}^{(t)}) \right)^2 + L_i D_{\mathcal{X}_i} \sum_{t=1}^{T} \epsilon^{(t)} + \frac{1}{2}\eta \sum_{t=1}^{T} \|\boldsymbol{u}_i(\boldsymbol{w}_{-i}^{(t)}) - \boldsymbol{u}_i(\boldsymbol{x}_{-i}^{(t)})\|_2^2.$$

*In particular, if $\eta := \frac{1}{2L}$ and $\epsilon^{(t)} \leq \frac{D_{\mathcal{X}_i}}{t^2}$ for any $t \in \llbracket T \rrbracket$,*

$$\mathrm{Reg}_i^{(T)} \leq 3LD_{\mathcal{X}_i}^2 - \frac{1}{8LD_{\mathcal{X}_i}^2} \sum_{t=1}^{T} \left( \mathrm{BRGAP}_i(\boldsymbol{x}^{(t)}) \right)^2 + \frac{1}{2}\eta \sum_{t=1}^{T} \|\boldsymbol{u}_i(\boldsymbol{w}_{-i}^{(t)}) - \boldsymbol{u}_i(\boldsymbol{x}_{-i}^{(t)})\|_2^2.$$

*Proof.* By Lemma A.20, it follows that for any $t \in \llbracket T \rrbracket$,

$$\frac{1}{2\eta}\|\boldsymbol{x}_i^{(t)} - \boldsymbol{x}_i^{(t-1)}\|_2^2 \geq \frac{\eta}{4D_{\mathcal{X}_i}^2} \left( \mathrm{BRGAP}_i(\boldsymbol{x}^{(t)}) \right)^2 - \frac{1}{2}\eta \|\boldsymbol{u}_i(\boldsymbol{w}_{-i}^{(t)}) - \boldsymbol{u}_i(\boldsymbol{x}_{-i}^{(t)})\|_2^2.$$

Summing over all $t \in \llbracket T \rrbracket$ and combining with Theorem A.19 implies the statement since $L_i D_{\mathcal{X}_i} \sum_{t=1}^{T} \epsilon^{(t)} \leq L_i D_{\mathcal{X}_i}^2 \sum_{t=1}^{T} \frac{1}{t^2} \leq 2LD_{\mathcal{X}_i}^2$. $\qquad\square$

We are now ready to prove Theorem 4.1, which is recalled below.

**Theorem 4.1.** *Suppose that all players are updating their strategies using CGD with $\epsilon^{(t)} \leq \frac{\min_i D_{\mathcal{X}_i}}{t^2}$ and learning rate $\eta = \frac{1}{2L}$ in a $(\lambda, \mu)$-smooth game $\mathcal{G}$, where $L$ is the Lipschtz-continuity parameter of $F$. Then, for any $\epsilon_0 > 0$ and $T \geq \frac{64L^2 D_{\mathcal{X}}^4}{\epsilon_0^2}$ iterations,*

1. *the average correlated distribution of play is a $\frac{4LD_{\mathcal{X}}^2}{T} - CCE$;*

2. *there is a time $t^{\star} \in \llbracket T \rrbracket$ such that*

$$\mathrm{SW}(\boldsymbol{x}^{(t^{\star})}) \geq \sup_{\epsilon \geq \epsilon_0} \min \left\{ \rho_{\mathcal{G}}(\lambda, \mu) \cdot \mathrm{OPT}_{\mathcal{G}} + \frac{\epsilon^2}{16(\mu+1)LD_{\mathcal{X}}^2}, \mathrm{PoA}_{\mathcal{G}}^{\epsilon} \cdot \mathrm{OPT}_{\mathcal{G}} \right\}. \tag{3}$$

*Proof.* First, by Corollary A.21,

$$\sum_{i=1}^{n} \text{Reg}_i^{(T)} \le 3L \sum_{i=1}^{n} D_{\mathcal{X}_i}^2 - \frac{1}{8LD_{\mathcal{X}}^2} \sum_{i=1}^{n} \sum_{t=1}^{T} \left( \text{BRGAP}_i(\boldsymbol{x}^{(t)}) \right)^2 + \frac{1}{2}\eta \sum_{t=1}^{T} \sum_{i=1}^{n} \|\boldsymbol{u}_i(\boldsymbol{w}_{-i}^{(t)}) - \boldsymbol{u}_i(\boldsymbol{x}_{-i}^{(t)})\|_2^2$$

$$= 3LD_{\mathcal{X}}^2 - \frac{1}{8LD_{\mathcal{X}}^2} \sum_{i=1}^{n} \sum_{t=1}^{T} \left( \text{BRGAP}_i(\boldsymbol{x}^{(t)}) \right)^2 + \frac{1}{2}\eta \sum_{t=1}^{T} \|F(\boldsymbol{w}^{(t)}) - F(\boldsymbol{x}^{(t)})\|_2^2$$

$$\le 3LD_{\mathcal{X}}^2 - \frac{1}{8LD_{\mathcal{X}}^2} \sum_{i=1}^{n} \sum_{t=1}^{T} \left( \text{BRGAP}_i(\boldsymbol{x}^{(t)}) \right)^2 + \frac{1}{2}\eta L^2 \sum_{t=1}^{T} \|\boldsymbol{w}^{(t)} - \boldsymbol{x}^{(t)}\|_2^2$$

$$\le 3LD_{\mathcal{X}}^2 - \frac{1}{8LD_{\mathcal{X}}^2} \sum_{i=1}^{n} \sum_{t=1}^{T} \left( \text{BRGAP}_i(\boldsymbol{x}^{(t)}) \right)^2 + \frac{1}{2}\eta L^2 \sum_{t=1}^{T} (\epsilon^{(t)})^2$$

$$\le 4LD_{\mathcal{X}}^2 - \frac{1}{8LD_{\mathcal{X}}^2} \sum_{i=1}^{n} \sum_{t=1}^{T} \left( \text{BRGAP}_i(\boldsymbol{x}^{(t)}) \right)^2, \tag{38}$$

where we used the fact that $\sum_{t=1}^{T}(\epsilon^{(t)})^2 \le D_{\mathcal{X}}^2 \sum_{t=1}^{T} \frac{1}{t^4} \le 2D_{\mathcal{X}}^2$ and $\eta = \frac{1}{2L}$. As a result, Item 1 follows directly from (38) by invoking the well-known fact that the CCE gap is bounded by $\max_{1 \le i \le n} \text{Reg}_i^{(T)}$.

For Item 2, let us fix any $\epsilon \ge \epsilon_0$. Suppose that at every time $t \in [\![T]\!]$ it holds that $\text{BRGAP}_i(\boldsymbol{x}^{(t)}) > \epsilon$ for some player $i \in [\![n]\!]$. Then, by (38) we conclude that

$$\sum_{i=1}^{n} \text{Reg}_i^{(T)} \le 4LD_{\mathcal{X}}^2 - \frac{1}{8LD_{\mathcal{X}}^2}\epsilon^2 T \le -\frac{1}{16LD_{\mathcal{X}}^2}\epsilon^2 T,$$

since $T \ge \frac{64L^2 D_{\mathcal{X}}^4}{\epsilon_0^2} \ge \frac{64L^2 D_{\mathcal{X}}^4}{\epsilon^2}$. As a result, by $(\lambda, \mu)$-smoothness, it follows that

$$\sum_{i=1}^{n} \text{Reg}_i^{(T)} \ge \lambda \text{OPT}_{\mathcal{G}} T - (1+\mu) \sum_{t=1}^{T} \text{SW}(\boldsymbol{x}^{(t)}),$$

in turn implying that

$$\frac{1}{T} \sum_{t=1}^{T} \text{SW}(\boldsymbol{x}^{(t)}) \ge \rho_{\mathcal{G}}(\lambda, \mu) \cdot \text{OPT}_{\mathcal{G}} - \frac{1}{T} \sum_{i=1}^{n} \text{Reg}_i^{(T)} \ge \rho_{\mathcal{G}}(\lambda, \mu) \cdot \text{OPT}_{\mathcal{G}} + \frac{\epsilon^2}{16(\mu+1)LD_{\mathcal{X}}^2}.$$

As a result, there is a time $t^\star \in [\![T]\!]$ such that $\text{SW}(\boldsymbol{x}^{(t^\star)}) \ge \rho_{\mathcal{G}}(\lambda, \mu) \cdot \text{OPT}_{\mathcal{G}} + \frac{\epsilon^2}{16(\mu+1)LD_{\mathcal{X}}^2}$. In the contrary case, if there is a time $t^\star \in [\![T]\!]$ such that $\text{BRGAP}_i(\boldsymbol{x}^{(t^\star)}) \le \epsilon$ for any player $i \in [\![n]\!]$, it follows that $\text{SW}(\boldsymbol{x}^{(t^\star)}) \ge \text{PoA}_{\mathcal{G}}^{\epsilon} \cdot \text{OPT}_{\mathcal{G}}$ (by definition). This concludes the proof. $\square$

In particular, we note that Corollary 4.3 stated earlier in the main body is an immediate consequence of Theorem 4.1 under Condition 4.2 since $\text{rPoA}_{\mathcal{G}} \ge \rho_{\mathcal{G}}$. It is also worth noting that for the special case of normal-form games, one can state Theorem 4.1 so that the first term in the right-hand side of (3) reads $\rho_{\mathcal{G}}(\lambda, \mu) \cdot \text{OPT}_{\mathcal{G}} + \frac{\epsilon^2}{32(\mu+1)L}$; thus, for a broad class of games (see Lemmas A.7 to A.9), the improvement over the smoothness bound is an absolute constant when $\epsilon$ and $\mu$ are also bounded by absolute constants. Relatedly, we should note that Theorem 6.2 applies in the regime where $\mu$ is polynomially bounded (we are not aware of any application of smoothness where this is not the case).

*Remark* A.22. It is direct to see that Item 2 of Theorem 4.1 can be refined so that there is a set $S \subseteq [\![T]\!]$, with $|S| \ge (1-\gamma)T$, so that

$$\frac{1}{|S|} \sum_{t \in S} \text{SW}(\boldsymbol{x}^{(t)}) \ge \sup_{\epsilon \ge \epsilon_0} \min \left\{ \rho_{\mathcal{G}} \cdot \text{OPT}_{\mathcal{G}} + \frac{\gamma \epsilon^2}{16(\mu+1)LD_{\mathcal{X}}^2}, \text{PoA}_{\mathcal{G}}^{\epsilon} \cdot \text{OPT}_{\mathcal{G}} \right\}.$$

*Remark* A.23. Theorem 4.1 can also be refined using the primal-dual framework of Nadav and Roughgarden [75] discussed in Appendix A.3, with the caveat that the variables $\{z_i\}_{i=1}^{n}$ need to

again have a bounded pairwise ratio. It is interesting to note, however, that using CGD enables making non-trivial conclusions even when a subset of the variables $\{z_i\}_{i=1}^n$ are zero in an optimal solution. In particular, when the quantity $\sum_{i=1}^n z_i \text{Reg}_i^{(T)}$ is nonnegative, we can readily draw the non-trivial conclusion that all players in the set $\{i \in [\![n]\!] : z_i \neq 0\}$ are eventually best responding (by virtue of Corollary A.21). It is not at all clear if such conclusions apply to OGD as well. In other words, using CGD one can essentially replace $\text{PoA}_{\mathcal{G}}^\epsilon$ in Theorem 4.1 by the price of anarchy with respect to a solution concept in which a *particular subset* of players are best responding. Shedding light to this type of guarantee necessitates understanding the implications of having some of the variables $\{z_i\}_{i=1}^n$ being zero in an optimal solution of the LP (21). Relatedly, even if we constraint $z_1 = z_2 = \cdots = z_n = z$, can we characterize the games for which $z \approx 0$? We explained earlier in Remark A.10 that such pathologies (typically) occur in constant-sum games, for which questions concerning social welfare are trivial.

### A.10 Beyond smoothness

In this subsection, we discuss an example in which the welfare predicted by the smoothness framework is far from the welfare obtained by learning algorithms such as OGD. We then suggest a natural direction that enables obtaining sharper predictions; we have already seen refined guarantees beyond the standard smoothness framework in Appendices A.3 and A.4, but here we explore a different direction.

Our example is again based on Shapley's game, a bimatrix game in normal form defined with the matrices

$$\mathbf{A} = \begin{bmatrix} 0 & 0 & 1 \\ 1 & 0 & 0 \\ 0 & 1 & 0 \end{bmatrix}, \mathbf{B} = \begin{bmatrix} 0 & 1 & 0 \\ 0 & 0 & 1 \\ 1 & 0 & 0 \end{bmatrix}. \tag{39}$$

**Claim A.24.** *Let $\mathcal{G}$ be the bimatrix game defined in (39). Then, $\text{rPoA}_{\mathcal{G}} = 0$.*

*Proof.* Let $(\boldsymbol{x}_1^\star, \boldsymbol{x}_2^\star) \coloneqq ((1,0,0),(0,1,0))$ be a welfare-maximizing joint action. By symmetry, the following argument will apply to any welfare-maximizing joint action. Consider $(\boldsymbol{x}_1, \boldsymbol{x}_2) = ((0,1,0),(0,1,0))$. Then, we have that $u_1(\boldsymbol{x}_1^\star, \boldsymbol{x}_2) + u_2(\boldsymbol{x}_2^\star, \boldsymbol{x}_1) = (\boldsymbol{x}_1^\star)^\top \mathbf{A} \boldsymbol{x}_2 + \boldsymbol{x}_1^\top \mathbf{B} \boldsymbol{x}_2^\star = 0 + 0$. As a result, the smoothness constraint corresponding to $(\boldsymbol{x}_1, \boldsymbol{x}_2)$ necessitates that $0 \geq \lambda$, which is incompatible with the constraint that $\lambda > 0$ (Definition 2.1). $\square$

In spite of the fact that $\text{rPoA}_{\mathcal{G}} = 0$, we see in Figure 2 (left) that OGD approaches close to the optimal social welfare 1. This can be explained by the fact that certain joint actions—in particular, those in the diagonal—are played with small probability under OGD (right-side of Figure 2). This motivates introducing a more refined notion of smoothness in which the smoothness constraints are only enforced on the joint actions that are visited with a non-negligible probability.

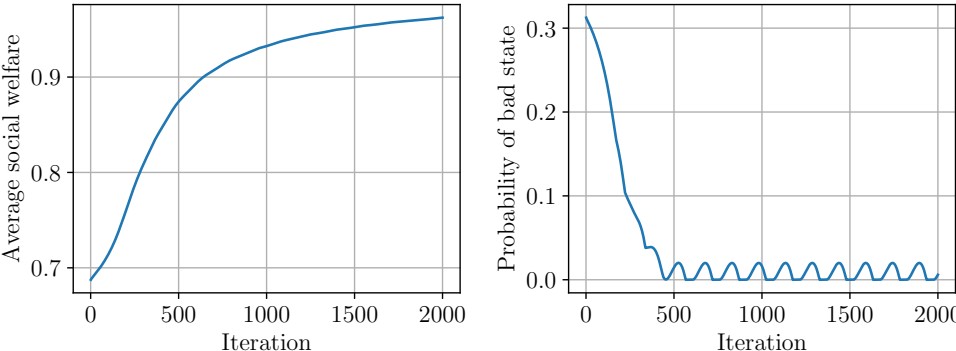

Figure 2: The behavior of OGD in the bimatix game (39) with $\eta \coloneqq 0.01$. On the left, we plot the average social welfare of the dynamics. On the right, we plot the probability of playing a joint action in the diagonal.

One class of games in which one can easily make such refinements concerns games solvable under iterated elimination of strictly dominated actions. As a concrete example, let

$$\mathbf{A} = \begin{bmatrix} 0 & 0 \\ 1 & 1 \end{bmatrix}, \mathbf{B} = \begin{bmatrix} 1 & 0 \\ 0 & 1 \end{bmatrix}. \tag{40}$$

**Observation A.25.** *Let $\mathcal{G}$ be the game defined in* (40). *Then,* $\mathrm{rPoA}_{\mathcal{G}} = \frac{1}{2}$. *Furthermore, if $\mathcal{G}'$ is the game[4] resulting from iterative removal of strictly dominated actions, then* $\mathrm{rPoA}_{\mathcal{G}'} = 1$.

---

[4]Under elimination of strictly dominated actions, this is always uniquely defined.

