# OpenReview forum: "On the Interplay between Social Welfare and Tractability of Equilibria"
_NeurIPS.cc/2023/Conference — NeurIPS 2023 poster_

### Official Review · Reviewer_Qpnv · 2023-06-17

**Soundness:** 3 good
**Presentation:** 2 fair
**Contribution:** 2 fair
**Rating:** 4
**Confidence:** 3

**Summary:**

This paper considers a specific type of smooth game first introduced in [Roughgarden 2015]. The main result of this paper is that when the game has robust PoA lower bounded by 1-$\epsilon$ (defined by the concept of smoothness of the game) and all the players apply optimistic gradient descent algorithm (OGD), the best iterate over the learning process will achieve $(\sqrt{\epsilon}/\sqrt{\delta n}, \delta)$ weak Nash-equilibrium, meaning that there are $\delta$ fraction of the players having low NE gap. The authors also show that under certain condition of the game, CGD, a variant of OGD, performs better social welfare guarantees.

**Strengths:**

- The paper considers a certain type of game with good rPoA guarantee and shows that classic OGD algorithm performs good results with provable guarantees.
- The general idea of the proof is intuitive and clear to me. The analysis also looks correct to me in general.


**Weaknesses:**

- One issue is the motivation of the study and the novelty of the proposed analysis. Specifically, it is less clear why the class of game with $rPoA\geq 1-\epsilon$ is an interesting class of games. It seems to me that this is more about a technical assumption that makes the proof work. More concretely, with respect to the first result showing the convergence to Nash, the analysis follows the recent line of works about last-iterate convergence and constant individual regret bound in general-sum games using OGD/OFTRL, which show that lower bounded sum-of-regret leads to stability of the dynamic, which furthre leads to low NE gap guarantees. The rPoA condition basically guarantees that the sum-of-regret is lower bounded according to the smoothness condition of the game.
- Another related issue is the trackability of rPoA. As mentioned mentioned by the authors, while rPoA can be calculated within polynomial time if the game is explicitly represented, meaning that the game representation already takes exponential space with respect to the number of players, which may not be the case when the utility function for each player is define by certain concrete functions. Therefore, deciding whether rPoA is lower bounded or not is also computationally intractable when the number of players is large.
- The description of Observation 3.3 does not look very clear. Specifically, the equivalence should define a certain choice of $(\lambda, \mu)$ in the definition of smooth game and I do not find a proof for this observation.
- The notations used in this paper are not very clear.
    - In section 3, many notations are with superscript $n$, which would be better if they are replaced by $*^{(n)}$ in order to distinguish with the exponent notation. Specifically, in line 194, $rPoA\geq 1-\epsilon^n$ is very misleading.
    - In line 302, should $rPoA_G^{\epsilon}$ be replaced by $PoA_G^{\epsilon}$, since $rPoA$ is not defined with respect to NE.
    - In line 324, the guarantee should be $rPoA_G\cdot OPT$. Specifically, in line below line 926, the RHS should also include $OPT$.

**Questions:**

- Can the authors explain more on the class of game with $rPoA\geq 1-\epsilon$? Is there a certain class of previously studied games or real applications that satisfy this condition?
- The obtained results for good NE gap strategy is with respect to the best-iterate. I wonder whether the results may also extend to the average-iterate case, which may lead to less computational complexity to derive the output strategy?


**Limitations:**

See weakness and questions for details.

---

> ### Author Rebuttal · Authors · 2023-08-08
>
> We are grateful to the reviewer for their feedback. Below we address the concerns.
>
> *“Specifically, it is less clear why the class of game with $rPoA≥1−\epsilon$  is an interesting class of games. [...] Is there a certain class of previously studied games or real applications that satisfy this condition?”*
>
> We believe that the main application of this condition is on large games. Indeed, two rather recent papers [37,22] show that $rPoA \to 1$ as $n \to \infty$ under very broad conditions, encompassing Walrasian auctions and Fisher markets, as well as general combinatorial auctions under probabilistic demand. Those are some of the most well-studied settings in algorithmic game theory, so the class of games in which $rPoA \geq 1 - \epsilon$ certainly contains previously studied games in the literature. Another application of this condition we provide in our paper is on games with bounded influence (see Corollary 3.4), which is another class of games with rich history and real applications.
>
> Furthermore, beyond large games and games with bounded influence, another class that satisfies this property are zero-sum games for which the Minty property holds (Observation 3.3), which is a classical condition in optimization. We believe that this connection is interesting as it connects two separate lines of work, and it allows us to use insights from the line of work on smoothness to better understand the Minty property, and vice versa; several such implications are highlighted in Lines 70-85.
>
> *“Another related issue is the trackability of rPoA. As mentioned mentioned by the authors, while rPoA can be calculated within polynomial time if the game is explicitly represented, meaning that the game representation already takes exponential space with respect to the number of players, which may not be the case when the utility function for each player is define by certain concrete functions. Therefore, deciding whether rPoA is lower bounded or not is also computationally intractable when the number of players is large.”*
>
> First, as we explained in our response above, there are many interesting classes of games where we know that $rPoA \to 1$, without requiring one to exactly compute rPoA.
>
> Second, and more importantly, the tractability of $rPoA$ is by no means necessary to our approach. Indeed, notice that executing the algorithm—which is of course computationally efficient even in multi-player games with a succinct representation—gives itself a sufficient and computationally efficient condition, without having to compute $rPoA$. Specifically, by running the algorithm and computing the (minimum) best response gap (which can be computed easily), there are two possibilities:
>
> i) if the best response gap is large, then our theory implies that $rPoA$ must be far from 1, in which case our theory is not applicable;
>
> ii) if the best response gap is small, we are close to Nash equilibrium, which is precisely what we were looking for in the first place.
>
> So computing $rPoA$ is not needed at all for our approach.
>
> *“The description of Observation 3.3 does not look very clear. Specifically, the equivalence should define a certain choice of $(\lambda,\mu)$ in the definition of smooth game “*
>
> This observation follows directly from the definitions, which is why a proof has not been included. Notice that a game being smooth by definition means that there exists a legitimate pair of finite parameters $(\lambda, \mu)$ that satisfies the smoothness property, which is why defining such a pair in Observation 3.3 is not needed. Indeed, the existence of any such pair suffices for the claimed equivalence.
>
> *“In section 3, many notations are with superscript $n$. which would be better if they are replaced by...”*
>
> We thank the reviewer for the suggestion. We will follow the reviewer’s recommendation in the revised version, as we see how our current notation can cause confusion.
>
> *“In line 302, should $rPoA_\mathcal{G}^\epsilon$ be replaced with $PoA_\mathcal{G}^\epsilon$ ?”*
>
> Indeed, this is a typo, thank you for pointing it out.
>
> *“In line 324, the guarantee should be [...]”*
>
> We stated Theorem 4.2 under the (normalizing) assumption that $OPT_{\mathcal{G}} = 1$ (as noted in Line 915), but we will follow the reviewer’s suggestion as we see how our current choice causes confusion.
>
> *“The obtained results for good NE gap strategy is with respect to the best-iterate. I wonder whether the results may also extend to the average-iterate case, which may lead to less computational complexity to derive the output strategy?”*
>
> Indeed, our main result can be extended to apply for an average iterate, not just the best iterate. Specifically, by selecting an iterate at random, there is a high probability that it will have a small equilibrium gap. We will make sure to point this out in the revised version.

---

> > ### Author Response · Authors · 2023-08-18
> >
> > We thank again the reviewer for their time and the helpful feedback. Given that the discussion period soon comes at an end, we wanted to see if our response adequately addressed the concerns, and if the reviewer has any further questions for us.

---

### Official Review · Reviewer_vrCA · 2023-06-26

**Soundness:** 3 good
**Presentation:** 3 good
**Contribution:** 3 good
**Rating:** 7
**Confidence:** 4

**Summary:**

This paper studies the convergence properties of no-regret dynamics in $(\lambda,\mu)$-smooth games. More precisely the authors show that for any $(\lambda,\mu)$-smooth game at which the bound of $\frac{\lambda}{1+\mu}$ converges to $1$ as the number of agents grows, the resulting Online Gradient Descent dynamics converges in polynomial time to an $(\epsilon,\delta)$-weak Nash Equilibrium. In an $(\epsilon,\delta)$-weak NE, $(1-\delta)$-fraction of the agents cannot increase their payoff by more that $\epsilon$ by deviating to another mixed strategy. The latter implies that if $\frac{\lambda}{1-\mu} = 1$ then the resulting OGD dynamics converges to an $\epsilon$-MNE in $O(1/\epsilon^2)$ steps. The latter reveals an interesting phase transition on the computational complexity of equilibrium computation of general multi-player games with PoA = 1 and the smaller class of $(\lambda,\mu)$-smooth games with $\frac{\lambda}{1-\mu} = 1$ (this naturally implies PoA = 1).

Motivated by the work of Roughgarden et al. establishing that no-regret dynamics in $(\lambda,\mu)$-smooth games guarantees $\frac{\lambda}{1-\mu}$ fraction of the optimal payoff, the authors show that the latter result can be improved for a specific class of $(\lambda,\mu)$-smooth satisfying an additional condition (Condition 4). More precisely, the authors show that if all agents adopt Clairvoyant Gradient Descent then the time-average joint probability distribution converges to a CCE while there exists an iteration at which the respective mixed strategy profile attains strictly more than $\frac{\lambda}{1+\mu}$ fraction of the optimal payoff.

**Strengths:**

I believe that the convergence results to an $(\epsilon,\delta)$-weak NE in $(\lambda,\mu)$-smooth games is a solid contribution. I also find surprising the phase transition on the computational complexity of equilibrium computation of games with PoA = 1 (that are PPAD hard) and the respective complexity of $(\lambda,\mu)$-smooth games with $\frac{\lambda}{1+\mu} = 1$. Also I find the second presented result, improving upon the $\frac{\lambda}{1+\mu}$ fraction of the optimal social welfare, interesting. I overall believe that this is a good paper aligned with the interest of the online learning/game theory audience of NeuRIPS.

**Weaknesses:**

In my opinion the only weakness of the paper concerns the comparison of the second result with the respective previous result of Roughgarden et al. As far as I understand, Roughgarden et al. show that the time-average payoff attained by any no-regret dynamics (in an $(\lambda,\mu)$-smooth game) is at least $\frac{\lambda}{1+ \mu}$ fraction of the optimal social welfare. However Theorem~4.2 guarantees social welfare strictly more than $\frac{\lambda}{1+ \mu}$ fraction the optimal one only at a specific iteration.

That being said I still consider the result interesting. However in case I am not missing something, an additionally discussion is needed so as to fairly compare the two results.

**Questions:**

Your second results refers to the $best-iterate$ while the result of Roughgarden et al. refets to the time-average payoff. Could you elaborate more on how the two results compare?

---

> ### Author Rebuttal · Authors · 2023-08-08
>
> We are grateful to the reviewer for their feedback.
>
> *“Theorem 4.2 guarantees social welfare strictly more than $\frac{\lambda}{1 + \mu}$  fraction the optimal one only at a specific iteration.”*
>
> While we stated Item 2 of Theorem 4.2 for a single iteration, it is direct to extend our proof so that the improvement in fact holds for almost all iterates (say 99% of them), not just a single one. More precisely, we can state Item 2 of Theorem 4.2 so that a $1- \delta$ fraction of the iterations has an average social welfare strictly better than rPoA by an additive factor of $\delta \cdot \epsilon_0^2 C(\mathcal{G})$. We will highlight this stronger result in the revised version.

---

> > ### Comment · Reviewer_vrCA · 2023-08-11
> >
> > Thank you for your response. I have read the other reviews and I am confident for my positive evaluation of the paper.

---

### Official Review · Reviewer_SmPo · 2023-06-27

**Soundness:** 3 good
**Presentation:** 3 good
**Contribution:** 3 good
**Rating:** 7
**Confidence:** 3

**Summary:**

This paper discusses the convergence and welfare of learning algorithms in smooth games. The authors show that when approximate full efficiency can be guaranteed via a smoothness argument, Nash equilibria are approachable under a family of no-regret learning algorithms, thereby guaranteeing fast and decentralized computation. They also leverage this connection to obtain new convergence results in large games, as well as extensions to Bayesian mechanisms. The paper unifies recent works in two-player zero-sum games, illuminating an equivalence between smoothness and a well-studied condition in the optimization literature known as the Minty property. Finally, the authors establish that a family of no-regret learning dynamics outperforms the welfare predicted by the smoothness framework under a generic condition, while at the same time guaranteeing convergence to the set of coarse correlated equilibria.

**Strengths:**

The strength of this paper lies in its contribution to the understanding of the convergence and welfare of learning algorithms in smooth games. The authors provide new insights into the relationship between efficiency and the behavior of a family of no-regret learning algorithms, and they also unify recent works in two-player zero-sum games.

**Weaknesses:**

The paper does have quite a few notations, but it's understandable. I've got a few questions, which might also point out some areas in the paper that could be improved. I've put these questions and potential weaknesses together in the question section for easy reference. But, I think this is an overall well-written paper.



**Questions:**

1. Is there any counterexample that if rPOG does not converge to 1, then the decentralized algorithm (OGD) provided in this paper does not converge to Nash Equilibrium? or If Condition 4.1 does not hold, then still thm 4.2.1 hold?
2. What is the indication of Condition 4.1? Are there some examples that condition 4.1 holds?




**Limitations:**

The authors adequately addressed the limitation and potential negative societal impact on their work.

---

> ### Author Rebuttal · Authors · 2023-08-08
>
> We are grateful to the reviewer for their feedback.
>
> *“Is there any counterexample that if $rPoA$ does not converge to 1, then the decentralized algorithm (OGD) provided in this paper does not converge to Nash Equilibrium?”*
>
> Yes. As we point out in Lines 272-274, there is a bimatrix game (see Proposition A.10 in Appendix A.3) in which $rPoA = 0.125$ and OGD does not converge to a Nash equilibrium.
>
> “Is there any counterexample…If Condition 4.1 does not hold, then still thm 4.2.1 hold”
>
> We do not know of an example where Condition 4.1 does not hold but the conclusion of Thm 4.2 applies, but we imagine that it should be the case that such examples exist.
>
> “Are there some examples that condition 4.1 holds?”
>
> Condition 4.1 is quite generic, so we expect it to hold in general. More specifically, as we explain in the paper, there are two sufficient assumptions to satisfy Condition 4.1. The first one is that $PoA > rPoA$; given that $PoA$ gives the worst-case quality over a smaller set compared to $rPoA$, we expect this condition to hold. More concretely, Figure 1 in Appendix A.5 shows that such is the case in random normal-form games, while in the revised version we will also highlight that the condition $PoA > rPoA$ is satisfied for all benchmark extensive-form games studied in the literature. The second assumption is a mild continuity condition regarding the behavior of $PoA$ under approximate Nash equilibria; as noted by Roughgarden [81], this is standard in this line of work for otherwise the entire analysis of $PoA$ becomes problematic (in that the conclusions of the theory are not robust to arbitrarily small perturbations). So Condition 4.1 is very generic. We finally point out that Theorem A.18 provides a more general result that does not rely on Condition 4.1.

---

> ### Comment · Reviewer_SmPo · 2023-08-10
> **Thank you**
>
> Thank you for your response.

---

### Official Review · Reviewer_9HSU · 2023-07-04

**Soundness:** 3 good
**Presentation:** 3 good
**Contribution:** 4 excellent
**Rating:** 8
**Confidence:** 3

**Summary:**

This work studies the connections between the efficiency of Nash equilibria, as measured for example by social welfare, and tractability of computing these equilibria through efficient no regret learning algorithms. The authors provide the key insight that the smoothness framework introduced by Roughgarden, that is typically used to analyze price of anarchy, can be also leveraged to analyze the efficient convergence of optimistic gradient descent (OGD) to Nash equilibria. They use this insight to prove that if as the number of players increases, the robust price of anarchy goes to 1 without increasing the games Lipschitz constant too quickly, then OGD converges to weak Nash equilibria. This finding unifies previous findings on zero sum games and is connected to existing work on the Minty property. Even beyond OGD, the authors study the connections between convergence to CCE and social welfare for clairvoyant GD (CGD). They show for the first time that convergence to CCE and outperforming restricted price of anarchy can be combined through an efficient algorithm.

**Strengths:**

Overall I believe that the paper makes a very strong and original contribution. To the best of my knowledge, connections between efficiency of equilibria and tractability have not received a lot of attention. This is evidenced by the fact that relatively simple observations like 3.3 are commonly overlooked. In addition to the novelty of this works topic, the technical results are very extensive and non trivial.

**Weaknesses:**

The paragraph from 179 to 190 could benefit from expansion. In the common setting where we analyze a fixed game, one can always normalize the utilities to get a Lipschitz constant of 1. I have a sense that such a normalization would affect the value of $\epsilon$ in the general case thus we cannot always get $\gamma$ to be small for free. I am not sure though. Adding more intuition would help.

**Questions:**

See above for a question/suggestion.

**Limitations:**

No limitations applicable.

---

> ### Author Rebuttal · Authors · 2023-08-08
>
> We are grateful to the reviewer for their feedback.
>
>
> *“The paragraph from 179 to 190 could benefit from expansion.  In the common setting where we analyze a fixed game, one can always normalize the utilities to get a Lipschitz constant of 1”*
>
> It is indeed the case that we can appropriately normalize the utilities so that the Lipschitz constant is 1, but here we operate in the usual setting where a bound on the utilities is imposed on the $\ell_\infty$ norm; this is why the Lipschitz constant with respect to the $\ell_2$ norm can be as large as $\Theta(\sqrt{n})$. We will clarify this in the revised version.

---

> > ### Comment · Reviewer_9HSU · 2023-08-19
> > **Thanks for the clarification**
> >
> > Thank you for the clarification!

---

### Official Review · Reviewer_Q8mF · 2023-07-06

**Soundness:** 3 good
**Presentation:** 3 good
**Contribution:** 3 good
**Rating:** 6
**Confidence:** 3

**Summary:**

This work studies the connection between efficiency and computation tractability of equilibrium in smooth games. Its major finding is that optimistic mirror descent reaches a weak Nash equilibrium for large games satisfying the property that asymptotically smoothness can guarantee efficiency of equilibria. This result establishes the connection between smoothness arguments and the approachability of no-regret learning algorithms to Nash equilibria.


**Strengths:**

This work draws new insights on an interesting angle connecting two fundamental subjects in algorithmic game theory. Robust price of anarchy characterizes the fraction of optimal social welfare that smoothness argument can guarantee for every Nash equilibria. It turns out from this paper that robust PoA $\to 1$ also ensures the convergence of no-regret dynamics, which is non-trivial. The results also recover existing theories of the covergence of OMD in two-player zero-sum games.


**Weaknesses:**

The rates (e.g. $n^{-\alpha/3}$) are dependent on the number of players instead of the number of iterations, which is different from what we commonly see/expect. I am not sure how useful the rates are, for example, in large games, still a $n^{2/3}$ number of players may yield bad responses.


**Questions:**

What are some natural examples where the conditions in Thm 3.1 are satisfied and non-trivial rates are obtained?

Is there direct generalization to mean-field games?

**Limitations:**

The authors adequately addressed the limitations.

---

> ### Author Rebuttal · Authors · 2023-08-08
>
> We are grateful to the reviewer for their feedback.
>
> *“What are some natural examples where the conditions in Thm 3.1 are satisfied and non-trivial rates are obtained?”*
>
> The most simple example is when $rPoA = 1$, in which case Thm 3.1 (in particular see the more general version of Theorem A.2) yields a $1/\sqrt{T}$-Nash equilibrium after $O(T)$ repetitions. As we point out in Observation 3.3, this already captures the well-studied Minty property, which is satisfied, for example, in two-player zero-sum games.
>
> Furthermore, in the regime where $rPoA \to 1$, Corollary 3.4 shows that the conditions of Thm 3.1 are satisfied in the natural class of games with bounded influence, which includes voting games; see reference [51] for further applications.
>
>
> *“In large games, still a $n^{2/3}$  number of players may yield bad responses.”*
>
> This is true for a certain regime of $\gamma^n$, but as we show in Corollary A.3 our approach can also establish convergence to a Nash equilibrium--a point where *all* players are best responding.
>
> *“Is there direct generalization to mean-field games?”*
>
> This is an interesting question. To apply our results in the mean-field regime the key difficulty seems to be showing that the robust price of anarchy ($rPoA$) approaches $1$. There is some work in the literature that identifies natural classes of mean-field games for which $PoA \to 1$ (e.g., see Carmona et al. (Price of Anarchy for Mean Field Games)), so it is plausible that the same applies to $rPoA$ as well, but formalizing this would require further work.

---

> > ### Comment · Reviewer_Q8mF · 2023-08-11
> >
> > Thank you. After reading the rebuttal and other reviews I decide to keep my positive evaluation.

---

### Comment · Area_Chair_iJuA · 2023-08-11
**Some questions**

Dear authors,

Thank you for your timely responses.

I have now gone through your original submission, the reviewers' input, and your rebuttals. As we are now in the author-committee discussion phase, I would like to jumpstart an exchange on the following points:
- Regarding the point raised by Reviewer Q8mF that "$n^{2/3}$ players could be playing bad responses". I read your response regarding Corollary A.3 (which is only in the supplement if I'm not mistaken), but this is contingent on a strong condition for the game's rPoA. Could you comment when this condition is verified in practice? Do you have an example?
- Regarding the guarantee of Theorems A.2 and A.18 (the precise version of Theorems 3.1 and 4.2 in the main), and related to a point raised by Reviewer Qpnv. As far as I understand, the guarantee is that if OGD/CGD is run for $T$ iterations (with $T$ specified in the supplement), there is some time $t^\ast$ such that $x(t^\ast)$ is a weak approximate equilibrium. In order to stop the algorithm at $t^\ast$ and harvest the solution, you ostensibly assume that the full structure of the game is known, correct? If not, what is your stopping criterion for the algorithm?
- Related to the above: it is not clear if you are taking a centralized, offline viewpoint (whereby OGD/CGD is run by a central optimization device) or a distributed, online viewpoint (whereby OGD/CGD is run by the players, each with local gradient/payoff information). If it is the former, could you comment on the memory/storage requirements as $n\to\infty$? If it is the latter, could you comment on the communication requirements in order to harvest the algorithm's "best iterate" at $t^\ast$? [In other words: how would the players know that they are at a weak approximate equilibrium in order to stop the algorithm?]
- Is there any difference between CGD and the classical proximal point algorithm of Rockafellar ("*Monotone operators and the proximal point algorithm*", SIOPT, 1976)?

Regards,

The AC

---

> ### Author Response · Authors · 2023-08-11
> **Thank you for the Questions**
>
> We are grateful to the AC for the engagement.
>
> - A simple class of examples revolves around games with bounded influence. In particular, as noted by Kearns and Mansour [51], there are certain games (such as voting games under the plurality rule) in which the maximum influence of a player on its utility is bounded by $1/n$; Corollary 3.4 then implies that only a constant number of players (independent on $n$) could have a large best response gap. Beyond such games, the main result of [37] shows that rPoA converges to $1$ with a rate of $1/\sqrt{n}$ in a certain regime, but this rate can be improved depending on the parameterization of the underlying combinatorial auction (Theorem 18 of [37] exactly quantifies this), in which case Corollary A.3 directly kicks in.
>
> - We do not require that the full structure of the game is known. First, we point out that the guarantee of Theorem A.2 readily extends (with high probability) by taking a random joint strategy, not necessarily the best one (Eq. (9) in the proof of Theorem A.2 suffices to see this). Further, if we want a deterministic guarantee each player can compute locally its best response gap (which can be trivially done with gradient feedback in our setting without any further information), and then terminate the algorithm when the desired accuracy has been reached (without any other information about the game). Regarding Theorem 4.2 (and Theorem A.18), it is direct to see from our proof that Item 2 can be made to hold for a $1-\delta$ fraction of the iterates, not just a single one, where the improvement over rPoA now is an additive factor of $\delta \cdot \epsilon_0^2 C(\mathcal{G})$. We will make sure to highlight those refinements in the revised version.
>
> - We are taking a decentralized online viewpoint wherein each player updates its strategy using the gradient of its own utility, although we believe that our results are also interesting from a purely centralized perspective. In particular, from a decentralized standpoint, as we discussed above if we are satisfied with a probabilistic guarantee then taking a random joint strategy in the trajectory suffices without inducing any communication overhead. Alternatively, we would need a minimal amount of communication to make sure to extract the best joint strategy. From a centralized perspective, the running time and the memory of the underlying algorithm of course scale with the number of players $n$, but the dependence is polynomial in $n$, which is the best we can hope for.
>
>
> - The CGD algorithm can be indeed seen as an approximate version of the classical proximal point algorithm, as discussed in detail in the references [35, 77, 17]. More precisely, CGD revolves around a contraction mapping in order to approximate the proximal point algorithm. As shown by Piliouras et al. [77], CGD only requires gradient feedback of the utilities, and can be implemented as an uncoupled no-regret learning algorithm.

---

> > ### Comment · Area_Chair_iJuA · 2023-08-12
> >
> > Thanks for your input, but the core of my questions remains:
> > > As noted by Kearns and Mansour [51], there are certain games (such as voting games under the plurality rule) in which the maximum influence of a player on its utility is bounded by $1/n$.
> >
> > In this case (i.e., for the type of games considered by Kearns and Mansour), since $L^n$ grows as $\sqrt{n}$ in general, we would have $\gamma^n = \sqrt{n}$, i.e., $\alpha = 1/2$, so as many as $O(n^{5/6})$ players might be playing a "bad response" (I'm getting the $5/6$ exponent as $1-\alpha/3$ from L178, but the description in the paragraph preceding Corollary 3.4 left me in doubt). Am I missing something here?
> >
> > > The main result of [37] shows that rPoA converges to $1$ with a rate of $1/\sqrt{n}$ in a certain regime, but this rate can be improved depending on the parameterization of the underlying combinatorial auction (Theorem 18 of [37] exactly quantifies this), in which case Corollary A.3 directly kicks in.
> >
> > Just to be clear, since $L^n$ may grow as $\sqrt{n}$, if the rPoA has a $1/\sqrt{n}$ rate of convergence, Corollary A.3 cannot be directly invoked, right? In the combinatorial auction setup of [37], Corollary A.3 can only be invoked under the specific parameter schedule of Theorem 18, is this correct?
> >
> > > First, we point out that the guarantee of Theorem A.2 readily extends (with high probability) by taking a random joint strategy.
> >
> > How would random stopping be applied to a multi-player context? If you have $n$ players, each drawing a stopping time from $1,\dots,T$, the probability that they pick the same $t^\ast$ (which is what's needed for the random stopping trick to work in your case) is exponentially small in $n$.
> >
> > This is related to my question regarding the centralized / decentralized viewpoint: are agents taking the decisions themselves or is there a central coordinator?
> >
> > > If we want a deterministic guarantee each player can compute locally its best response gap (which can be trivially done with gradient feedback in our setting without any further information).
> >
> > If I'm not mistaken, Eq. (32) provides a guarantee for the sum of the players' best response gaps, and there is no guarantee that all players will reach, at the same time, a state where their individual best response gap is small, right? If so, how would a player know when to stop? I don't see a way of leveraging the PoA bound without summing over all players' regrets / gaps - am I wrong in this?
> >
> > In fact, if I'm not further mistaken, when the algorithm stops, there will still be a fraction of players that are not playing approximate best responses, so their stopping criterion would not have been activated. In this case, how would the players know "who is who" (i.e., who stops and who doesn't)?
> >
> > I could potentially see how this could be circumvented if players knew the game (and/or they were able to observe each other's mixed strategies, an assumption which is stronger than requiring access to a deterministic gradient oracle), but not in the current setting. Am I missing something?
> >
> > > Alternatively, we would need a minimal amount of communication to make sure to extract the best joint strategy.
> >
> > What does "minimal" mean here, exactly? If each player needs to communicate with every other player, we are talking about $O(n^2)$ bits exchanged, right?  And, if players are allowed to exchange information about their mixed strategies, aren't we talking about an information model which is more exacting than requiring access to a deterministic gradient oracle?
> >
> > > The CGD algorithm can be indeed seen as an approximate version of the classical proximal point algorithm.
> >
> > Understood. I misread $w^t$ in (21) as $x^t$, thanks for clarifying. In retrospect, it would have been much clearer if this had been included in the main, as the textual description can lead to misunderstandings.
> >
> > That being said, I still have a question: how would players run the fixed-point iteration (21) in a decentralized way? If each player has their own stopping criterion - say $\epsilon/n$ - then one player might stop updating prematurely, so it is not clear if the standard fixed-point analysis applies (at least not without knowledge of the game or knowledge of each other's strategies). Could you comment on that as well?
> >
> > Best,
> >
> > The AC

---

> > > ### Author Response · Authors · 2023-08-12
> > >
> > > Thank you for the follow-up questions.
> > >
> > > *“In this case (i.e., for the type of games considered by Kearns and Mansour), since $L^n$ grows as $\sqrt{n}$ in general...”*
> > >
> > > $L^n$ does not have to grow as $\sqrt{n}$ in such games. In particular,  suppose that each player $i$ selects $x_i \in [0, 1]$, so that $u_i^j(x_1, \dots, x_n) = u_i^j( \frac{\sum_{i'=1, i' \neq i }^n x_{i'} }{n})$ where $u_i^j : \mathbb{R} \to \mathbb{R}$ is a $1$-Lipschitz utility function corresponding to action $j = 0, 1$. Let us denote by $S = \frac{1}{n}  \sum_{i=1}^n x_{i}$. For two joint strategy profiles $x, x’$, we have
> > >
> > > $\sum_{i=1}^n \sum_{j =0}^1 | u_i^j(S) - u_i^j(S') |^2 \leq 2 \sum_{i=1}^n | S - S' |^2  \leq 2 \sum_{i=1}^n | x_i - x_i’  |^2$.
> > >
> > > So, $L^n$ is independent on $n$, which in turn implies that only a constant number of agents can have a large best response gap (this follows by taking $\delta = \Theta(1/n)$ in Eq. (2)), as we claimed. This same of course applies beyond taking the average as the summarization function. We apologize for not explaining this argument in more detail in our earlier response.
> > >
> > > Corollary A.3 cannot be directly invoked, right? In the combinatorial auction setup of [37], Corollary A.3 can only be invoked under the specific parameter schedule of Theorem 18, is this correct?
> > >
> > > Yes, this is correct.
> > >
> > > *“How would random stopping be applied to a multi-player context? If you have $n$ players, each drawing a stopping time from $1, \dots, T$, the probability that they pick the same $t^\star$ (which is what's needed for the random stopping trick to work in your case) is exponentially small in $n$.”*
> > >
> > > For this we are assuming that players have access to a common source of randomness. Alternatively, if there is no access to a common source of randomness, a central coordinator can choose a random index $t^\star$, and transmit $\log (\lceil 1/T \rceil )$ bits to each player.
> > >
> > > *“In fact, if I'm not further mistaken, when the algorithm stops, there will still be a fraction of players that are not playing approximate best responses, so their stopping criterion would not have been activated. In this case, how would the players know "who is who" (i.e., who stops and who doesn't)?”*
> > >
> > > Let us fix some parameters $\epsilon, \delta$, with the goal of terminating at an $(\epsilon, \delta)$-weak Nash equilibrium. At each iteration every player determines with its own local information whether its own best response gap is at most $\epsilon$, and transmits that information—a single bit of information—to a central coordinator. The coordinator then sums those bits, and if their fraction is above $1-\delta$ the coordinator signals to the players that they can terminate. No further information about the game is needed to implement this.
> > >
> > > *“I could potentially see how this could be circumvented if players knew the game (and/or they were able to observe each other's mixed strategies, an assumption which is stronger than requiring access to a deterministic gradient oracle), but not in the current setting. Am I missing something?”*
> > >
> > > As we explained above we do not need any knowledge of the game beyond a deterministic gradient oracle.
> > >
> > > *“What does "minimal" mean here, exactly? If each player needs to communicate with every other player, we are talking about $O(n^2)$ bits exchanged, right? And, if players are allowed to exchange information about their mixed strategies, aren't we talking about an information model which is more exacting than requiring access to a deterministic gradient oracle?”*
> > >
> > > Players do not have to exchange information about their mixed strategies, and they only need to observe gradient feedback. In particular, as we explained above, players only have to send a single bit in every iteration to a central coordinator, which amounts to $n$ bits of communication, not $O(n^2)$. We used the word “minimal” in our previous response because $1$ bits of information is negligible compared to the bit complexity of the gradient.
> > >
> > > *"That being said, I still have a question: how would players run the fixed-point iteration (21) in a decentralized way? If each player has their own stopping criterion - say $\epsilon/n$ - then one player might stop updating prematurely, so it is not clear if the standard fixed-point analysis applies (at least not without knowledge of the game or knowledge of each other's strategies). Could you comment on that as well?"*
> > >
> > > The stopping criterion could be that all players apply the contraction for a fixed number of iterations, which is the same for each player. As long as there is an agreed bound on that number of iterations one can use the standard fixed-point analysis.

---

> > > > ### Comment · Area_Chair_iJuA · 2023-08-13
> > > >
> > > > Thanks for the clarification regarding the Lipschitz constant in voting games.
> > > >
> > > > For the rest, in all your replies, a central coordinator is required to take an active part in the decision-making - e.g., to process the players' best response gaps and then send a signal to terminate the process (my comments concerned the case where no such entity is assumed). From a game-theoretic perspective, this hybrid model raises itself several questions (for example, truthfulness in reporting: players may have an incentive to misreport their gap or if, by so doing, they believe they can keep the game going and steer it to a better outcome), but I do not want to belabor the point.
> > > >
> > > > Thank you for your replies. Regards,
> > > >
> > > > The AC

---

> > > > > ### Author Response · Authors · 2023-08-14
> > > > >
> > > > > Truthfulness is an interesting consideration. While the proposed distributed protocol is indeed by no means truthful, lack of truthfulness is a general issue encountered under regularized learning algorithms. Although there have been some recent papers on exploitability of learning dynamics, most of the work does not address exploitability. So, while we are not addressing exploitability, neither is the bulk of this literature. We can include a discussion on this point if the AC considers it important for our setting.
> > > > >
> > > > > We thank the AC again for their time.

---

### Decision · Program_Chairs · 2023-09-21

**Decision:**

Accept (poster)

**Comment:**

This paper treats the problem of equilibrium convergence in large games whose price of anarchy goes to $1$ as the number of players grows to infinity. The authors assume that players have access to perfect payoff gradients and study the convergence of the optimistic gradient descent (OGD) algorithm. The paper's main result is that, if OGD is run by the players for a sufficiently large number of iterations, the history of play will be concentrated on states where most players have a minimal incentive to deviate. In addition, the authors study the players' social welfare and estimate its rate of convergence as a function of the game's robust price of anarchy.

For the most part, the reviewers appreciated the paper's technical contributions, the authors' rebuttal answered most of the committee's concerns, and there were no major objections to an "accept" decision. After my own discussion with the authors and the reviewers, I concur with this assessment, subject to the following clarifications / revisions for the camera-ready phase:
1. The statement of Theorems 3.1 and 4.2 (though mostly Theorem 3.1) should be made precise. The authors already included an informal version of their results in the introduction, so they should replace the current version of Theorems 3.1/4.2 with the statements A.2/A.18 from the appendix and avoid vague statements like "OGD reaches [some state]" .
2. The authors should clarify that, in addition to a perfect gradient oracle, the guarantees they provide for OGD require back-and-forth communication with a central coordinator to terminate the process. I would also recommend providing a more thorough description of how the algorithm is to be implemented in a decentralized setting, but I leave the details of this to the authors.
3. As pointed out by one of the reviewers, if the price of anarchy decreases at a moderate rate, a significant number of players could be playing a bad response at the end of the learning process. While this guarantee can be improved in certain cases, this is a limitation that the authors should discuss in more detail.

Subject to the above modifications (which amount to a minor revision of the paper), I am happy to recommend acceptance.